# Return of Frustratingly Easy Unsupervised Video Domain Adaptation

**Pengfei Wei** [1]   **Yiqun Sun** [1]   **Zhiqiang Xu** [2]   **Yiping Ke** [3]   **Lawrence B. Hsieh** [1]

## Abstract

Unsupervised video domain adaptation (UVDA) is a practical but under-explored problem. In this paper, we propose a frustratingly easy UVDA method, called *MetaTrans*. Specifically, *MetaTrans* adopts a concise learning objective that contains only two fundamental loss terms. Despite the simplicity of the learning objective, *MetaTrans* embodies an advanced UVDA idea, that is, handling the spatial and temporal divergence of cross-domain videos separately, through a subtle model architecture design. By implementing a temporal-static subtraction module, *MetaTrans* effectively removes spatial and temporal divergence. Extensive empirical evaluations, particularly on various cross-domain action recognition tasks, show substantial absolute adaptation performance enhancement and significantly superior relative performance gain compared with state-of-the-art UVDA baselines.

## 1. Introduction

Unsupervised video domain adaptation (UVDA) (Xu et al., 2022a) aims to transfer shared knowledge between different video domains. It is an under-explored research problem, but has immense potential in various video-based practical applications such as video conferencing, video editing, and video content analysis. Cross-domain action recognition, as one of the most popular UVDA tasks, has attracted increasing research attention over the past few years (Sahoo et al., 2021; da Costa et al., 2022a; Wei et al., 2023).

Image-based UDA methods (Tzeng et al., 2017) can be applied directly to the UVDA task but achieve unsatisfactory performance, as they only handle the spatial divergence of static frames while overlooking the temporal dependency between frames during adaptation. Some works (da Costa et al., 2022b; Reddy et al., 2024; Xu et al., 2024) have emerged specifically for UVDA putting emphasis on temporal alignment. The key idea is to reduce the cross-domain divergence by aligning frame-level and video-level features. In general, these methods do not distinguish the spatial and temporal domain divergence and handle them together.

Recently, researchers have begun to focus on reducing both the spatial and temporal divergence in UVDA. A *TranSVAE* framework (Wei et al., 2023) has been proposed to address UVDA by disentanglement. It assumes that a video is generated by static and dynamic latent factors, and proposes the simultaneous elimination of spatial and temporal divergence through a tailored static and dynamic latent feature disentanglement, specifically engineered for adaptation. In (Lin et al., 2024), a human encoder and a context encoder are developed to decouple the static context and the dynamic human motion, although it is specifically designed for the human action recognition task. These methods attain state-of-the-art adaptation results on several UVDA benchmark datasets, and remarkably they surpass some multi-modality methods by only using RGB features.

Despite the fact that these works have made strides in UVDA, we observe a potential bottleneck that may impede their practical applicability. They generally use complex models that incorporate multiple submodules, e.g., with up to 7 loss terms in *TranSVAE* and 5 loss terms in HCT in their final objective function. To obtain the optimal adaptation performance for a UVDA task, multiple training runs are needed to find the best weight for each loss. Although the authors propose to reduce the number of training runs by some tricks, they still need around ten thousand training runs for a single task if using greedy grid search. In this paper, our aim is to develop a UVDA method that employs a streamlined model structure while being capable of achieving state-of-the-art adaptation performance. Theoretical analysis of image-based UDA (Ben-David et al., 2006) indicates that adaptation performance improvements often originate from minimized source training risks and reduced domain divergence. This motivates us to aspire our method to be as simple as this foundational UDA work, comprising only two losses in the final objective function: the source supervision loss and the domain divergence minimization loss. By doing so, we can significantly reduce the number

[1]Magellan Technology Research Institute (MTRI), Japan [2]Mohamed bin Zayed University of Artificial Intelligence, UAE [3]Nanyang Technological University, Singapore. Correspondence to: Yiqun Sun <duke.sun@mtri.co.jp>.

*Proceedings of the $43^{rd}$ International Conference on Machine Learning*, Seoul, South Korea. PMLR 306, 2026. Copyright 2026 by the author(s).

of training runs required for the weight search.

However, simply using the two losses is insufficient to attain state-of-the-art UVDA performance. The key is to implement an advanced adaptation idea with this simple objective. In particular, we expect our model to not only remove the spatial divergence but also proceed with temporal alignment. Note that, instead of stacking multiple losses to do so, we adhere to the above two losses while achieving the idea by developing a temporal-static subtraction module. Specifically, we design a self-attention-based model structure to simultaneously produce a static and a temporal feature representation from input sequences. Meanwhile, we remove spatial domain divergence by subtracting the static feature representation from the temporal one, followed by eliminating temporal domain divergence based on the subtraction results. It is worth noting that our temporal-static subtraction module has a *permutation-invariant* nature, which is theoretically proved. Such permutation-invariance guarantees that the spatial domain divergence can be effectively removed by the simple subtraction. As a result, we present *MetaTrans*, a frustratingly easy UVDA method that achieves good adaptation performance with only two losses.

## 2. Related Work

UVDA is an under-explored problem, and a few methods have attempted UVDA. Chen *et al.* (Chen et al., 2019) proposed a temporal attentive adversarial adaptation network for frame- and video-level alignment. Instead, clip-level alignment, region-level alignment, and pixel correlation discrepancy alignment are explored in (Choi et al., 2020), (Hu & Zhu, 2023), and (Xu et al., 2022b), respectively. Wu *et al.* (Wu et al., 2022) further proposed mixing different levels of feature alignment. The attention mechanism is also widely used in UVDA. In (da Costa et al., 2022b), the authors directly applied an attention module to learn the domain-invariant features. The temporal co-attention network (Pan et al., 2020) adopted cross-domain attention on common frames for temporal alignment. In (Luo et al., 2020), the authors proposed modeling the cross-domain correlations using a bipartite graph network topology. The work (Broomé et al., 2023) studied the cross-domain robustness for both convolution and attention models.

Some studies emphasize the background. Sahoo *et al.* (Sahoo et al., 2021) developed an end-to-end temporal contrastive learning framework with background mixing and target pseudo-labels. In contrast, Lee *et al.* (Lee et al., 2024) proposed to learn temporal order sensitive representations and background invariant representations to remove background shifts. Other works improve the classifier to boost the adaptation performance. Chen *et al.* (Chen et al., 2022) learned a multi-head domain classifier for multi-level temporal attentive features to achieve better alignment, and

(da Costa et al., 2022a) exploited the combination of cross-entropy and contrastive losses to form a two-headed target classifier. In (Reddy et al., 2024), the authors apply self-training on masked target data and use the video student model and the image teacher model together for UVDA.

Unlike the methods discussed above that focus on the temporal alignment in UVDA, recent studies start to handle both the spatial and temporal divergence. Wei *et al.* (Wei et al., 2023) proposed a general framework that handles UVDA from a disentanglement perspective. Lin *et al.* (Lin et al., 2023) explicitly model spatial-temporal dependencies between video contents at multiple space-time scales using a clustering-like process. Recently, HCT (Lin et al., 2024) decouples human motion from the environment with a human encoder and a context encoder specifically for human action recognition tasks.

There are also works using multi-modality data for UVDA. Most of them use flow as an auxiliary input, e.g. (Munro & Damen, 2020; Song et al., 2021; Kim et al., 2021; Yang et al., 2022). In addition to flow, some methods also explored other auxiliary inputs, e.g., audios (Zhang et al., 2022), web images (Lin et al., 2022), and wild data (Yin et al., 2022). In recent work (Planamente et al., 2024), the authors combined RGB features, flows, and audios for UVDA. In this paper, we only use RGB features, and we take the extension to multi-modality as a promising future work direction.

Instead of working on UVDA, some work deals with UVDA variants, i.e. open-set UVDA and source-free UVDA. For open-set UVDA, the key is to reduce the adaptation effect of unknown samples. In (Chen et al., 2021), the authors used the boost strategy to align the confident samples. Wang *et al.* (Wang et al., 2021) separated known and unknown categories using a novel dual-metric discriminator. Similar to (Sahoo et al., 2021) for UVDA, Zara *et al.* (Zara et al., 2023b) proposed a contrastive learning framework that learns discriminative and well-clustered features for open-set UVDA. Recent work (Zara et al., 2023c) takes advantage of the pre-trained language and vision model (CLIP) that automatically generates target-private class names.

For source-free UVDA, the difficulty lies in the lack of source data during the adaptation stage. Dasgupta *et al.* (Dasgupta et al., 2025) utilized pseudo-labels and recursively corrected the pseudo-labels to fine-tune the source pre-trained model. Xu *et al.* (Xu et al., 2022c) extended the idea of (Pan et al., 2020) to source-free UVDA, while (Huang et al., 2022) combined the idea of (Chen et al., 2019) with multi-modal inputs. In (Li et al., 2023), the authors proposed using spatial and temporal augmentations in target videos to update the pre-trained source model. These UVDA variants are not our focus in this work, but we still discuss them for a comprehensive UVDA related work review.

## 3. The Proposed Method

### 3.1. Problem Setting

We consider the typical UVDA setting with one source domain $\mathcal{S}$ and one target domain $\mathcal{T}$. The source $\mathcal{S}$ has sufficient labeled data, i.e., $\{(\mathbf{X}_i^{\mathcal{S}}, y_i^{\mathcal{S}})\}_{i=1}^{N_{\mathcal{S}}}$. The target $\mathcal{T}$ has only unlabeled data, $\{\mathbf{X}_i^{\mathcal{T}}\}_{i=1}^{N_{\mathcal{T}}}$. For a domain $\mathcal{D} \in \{\mathcal{S}, \mathcal{T}\}$, $\mathbf{X}_i^{\mathcal{D}}$ has $T$ frames $\{\mathbf{x}_{i\_1}^{\mathcal{D}}, ..., \mathbf{x}_{i\_T}^{\mathcal{D}}\}$ in total, where $\mathbf{x}_{i\_t}^{\mathcal{D}}$ is the *t-th* image frame. We also denote the total amount of data $N = N^{\mathcal{S}} + N^{\mathcal{T}}$. The domains $\mathcal{S}$ and $\mathcal{T}$ have different data distributions but share the same label space. The objective is to utilize both $\{(\mathbf{X}_i^{\mathcal{S}}, y_i^{\mathcal{S}})\}_{i=1}^{N_{\mathcal{S}}}$ and $\{\mathbf{X}_i^{\mathcal{T}}\}_{i=1}^{N_{\mathcal{T}}}$ to train a good prediction model for the domain $\mathcal{T}$.

### 3.2. Streamlined Objective Function

In this paper, we aim to develop a UVDA method that achieves good adaptation performance while minimizing the number of training runs. Inspired by the theory of image-based UDA study (Ben-David et al., 2006), i.e., explicit minimization of domain divergence and source risk results in good domain adaptation representations, we enforce the learning objective of our model to incorporate only two fundamental losses: domain divergence minimization loss and source task supervision loss. These two losses are basically indispensable in every UDA or UVDA method, where the former is the primary mechanism of reducing domain gaps, while the latter is for the final prediction.

We implement the loss of domain divergence minimization using the domain adversarial idea (Wei et al., 2023). Specifically, we construct a domain classifier that is tasked with discerning whether the data originate from $\mathcal{S}$ or $\mathcal{T}$. This gives the following domain divergence minimization loss:

$$\mathcal{L}_{adv} = \mathbb{E}_i[\mathbb{E}_t[\mathcal{L}(C_d(\mathcal{M}(\mathbf{X}_i^{\mathcal{D}}), d_i)]], \tag{1}$$

where $\mathbb{E}[\cdot]$ is the expectation operator, $\mathcal{M}$ is our model, $\mathcal{L}$ is the cross-entropy loss, $C_d$ is the domain classifier and $d_i$ is the domain label.

The loss of task supervision applied to the source-labeled data is essential as it facilitates the training of the prediction model. Existing work (Chen et al., 2022) has highlighted the advantages of adding target pseudo-labeled data in task supervision loss. Thus, we also consider both. Specifically, in the training phase, we utilize the prediction network from the preceding epoch to generate target pseudo-labels. To ensure the reliability of these labels, we initially train the prediction network solely on the source supervision for several epochs. In sum, the final task-specific supervision loss is as follows.

$$\mathcal{L}_{cls} = \mathbb{E}_i[\mathcal{L}(\mathcal{M}(\mathbf{X}_i^{\mathcal{D}}), y_i^{\mathcal{D}})], \tag{2}$$

where $y_i^{\mathcal{D}}$ is the label (pseudo-label) of $\mathbf{X}_i^{\mathcal{S}}$ ($\mathbf{X}_i^{\mathcal{T}}$). The final objective function is the combination of the domain divergence minimization and the task supervision losses:

$$\mathcal{L}' = \mathcal{L}_{cls} + \lambda_1 \mathcal{L}_{adv}, \tag{3}$$

where $\lambda_1$ is the balance weight. With Eq. (3), we only need to search for the optimal value of $\lambda_1$, which greatly minimizes the number of training runs.

### 3.3. Model Architecture

Using only the learning objective of Eq. (3) is insufficient to achieve state-of-the-art UVDA performance, despite the significant reduction in the number of training runs. The next step is to explore how to design the structure of the model $\mathcal{M}$ to achieve advanced adaptation ideas while not introducing extra losses. State-of-the-art methods have shown the superiority of separately handling the spatial and temporal divergence, e.g. *TranSVAE* (Wei et al., 2023) disentangles input video sequences as static and dynamic latent factors using a VAE-based model, and then uses the dynamic latent factors for the UVDA task. However, effective disentanglement of static and dynamic latent factors requires employing up to 7 losses. *TranSVAE*, when equipped solely with task supervision and adversarial losses, yields only marginal improvement (1.58%) over the source-only baseline, considerably short of optimal results. Our aim is to design the architecture of $\mathcal{M}$ so that spatial and temporal divergences are handled separately at the model structural level, while retaining Eq. (3) as the learning objective.

Precisely, we develop a temporal-static subtraction module that simultaneously generates a temporal feature representation and a static feature representation. The spatial domain divergence is removed by simply subtracting the static feature representation from the temporal one. The losses, as outlined in Eq. (3), are acted on the subtraction results, further reducing the temporal divergence on the one hand and outputting the final prediction model on the other.

Note that the crux of the temporal-static subtraction module lies in generating good static and temporal feature representations, with particular emphasis on the former. Intuition with respect to a good static feature representation suggests that it should encapsulate global information and exhibit consistency across different video sequences. In particular, when considering a sequence, if we were to randomly shuffle the order of frames within this sequence, the static feature representation extracted from both the shuffled and original sequences should be equal or exhibit minimal discrepancy. Ideally, this condition should hold for any permutation of frame order within the sequence. This motivates us to develop a temporal permutation-invariant sub-module to learn the static features. To achieve this, we first introduce the definition of temporal permutation invariance.

**Definition 1.** Given $\mathbf{X} \in \mathbb{R}^{d \times T}$ where $T$ is the temporal dimension and $d$ is the number of features and denoted $\pi$

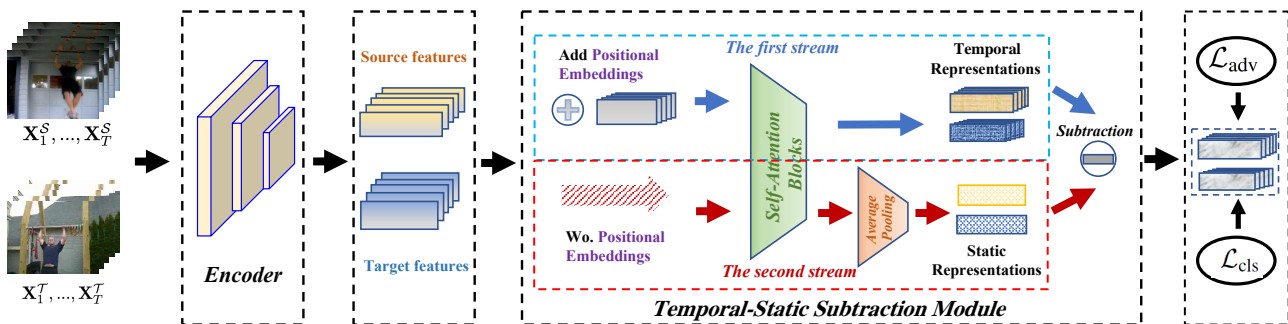

Figure 1. *MetaTrans* overview. The input videos are fed into an encoder to extract visual features, followed by a temporal-static subtraction module to learn a static representation and a temporal representation from the visual features without and with positional embeddings, respectively. A latent temporal embedding is obtained by subtracting the static features from temporal ones.

as a permutation of $T$ elements, we define a transformation $\mathcal{T}_\pi : \mathbb{R}^{d \times T} \to \mathbb{R}^{d \times T}$ a temporal permutation if $\mathcal{T}_\pi(\mathbf{X}) = \mathbf{X}\mathbf{P}_\pi$ where $\mathbf{P}_\pi \in \mathbb{R}^{T \times T}$ denotes the permutation matrix associated with $\pi$ defined as $\mathbf{P}_\pi = [\mathbf{e}_{\pi(1)}, \mathbf{e}_{\pi(2)}, ..., \mathbf{e}_{\pi(T)}]$ and $\mathbf{e}_{\pi(i)}$ is a one-hot vector of length $T$ with its $i$-th element being 1. An operator $\mathcal{O} : \mathbb{R}^{d \times T} \to \mathbb{R}^{d \times T}$ is temporally permutation invariant if $\mathcal{O}(\mathcal{T}_\pi(\mathbf{X})) = \mathcal{O}(\mathbf{X})$ for any $\mathbf{X}$ and any temporal permutation $\mathcal{T}_\pi$.

With the above definition, we devise an attention-based model architecture (Vaswani et al., 2017) for $\mathcal{M}$. Our $\mathcal{M}$ employs two streams $\mathcal{M}_1$ and $\mathcal{M}_2$: $\mathcal{M}_1$ utilizes sequences with positional embeddings to learn the temporal representation, while $\mathcal{M}_2$ uses sequences without positional embeddings to learn the static representation. In particular, for the first stream, video sequences initially pass through a positional embedding block to incorporate positional information. Subsequently, they traverse the attention blocks to reveal temporal information within the data. In contrast, for the second stream, video sequences directly undergo attention blocks, followed by an average operation to extract the static information embedded in the data. The framework overview of $\mathcal{M}$ is shown in Figure 1. The crucial insight lies in the temporal *permutation invariant* nature of $\mathcal{M}_2$, as shown in Theorem 1.

**Theorem 1.** *Consider $\mathcal{M}_2$ composed of: (i) multi-head self-attention without positional embeddings, (ii) residual connections, (iii) layer normalization performed per feature dimension, (iv) a feature-wise feed-forward network with shared parameters across time, and (v) an average operator. Specifically, letting $\mathbf{X}$ be the input sequence and $\mathcal{T}_\pi$ be any temporal permutation, we have: $\mathcal{M}_2(\mathcal{T}_\pi(\mathbf{X})) = \mathcal{M}_2(\mathbf{X})$.*

The full proof of Theorem 1 is stated in section A of the appendix. With Theorem 1, we can ensure that $\mathcal{M}_2$ is capable of generating an identical static feature representation, irrespective of any arbitrary change in frame order. It is worth noting that appending an average operation at the end of $\mathcal{M}_1$ also yields another form of static feature rep-

resentation, but this way lacks permutation invariance due to the incorporation of positional embeddings, which are inherently non-permutation invariant. This becomes evident because the addition of positional embeddings encodes positional information within the input sequence, rendering any shuffling operation meaningless.

Denoted $\mathbf{P}$ as positional embeddings, we remove the spatial domain divergence by subtracting the static latent feature representation from the temporal one:

$$\mathbf{F}_i^{\mathcal{D}} = \mathcal{M}_1(\mathbf{X}_i^{\mathcal{D}} + \mathbf{P}_i^{\mathcal{D}}) - \mathcal{M}_2(\mathbf{X}_i^{\mathcal{D}}). \qquad (4)$$

Note that the static features are repeated $T$ times to perform the subtraction. This results in a latent temporal embedding $\mathbf{F}_i^{\mathcal{D}}$. Subsequently, we incorporate Eq. (1) for frame-level and video-level temporal alignments, where the former is in $\mathbf{F}_i^{\mathcal{S}}$ and $\mathbf{F}_i^{\mathcal{T}}$ and the latter is in the video-level features obtained from the frame aggregation network ($\mathcal{FAN}$) widely used in the existing UVDA studies (Chen et al., 2019; Wei et al., 2023).

$$\mathcal{L}_{adv} = \mathbb{E}_i[\mathbb{E}_t[\mathcal{L}(C_d(\mathbf{F}_i^{\mathcal{D}}, d_i) + \mathcal{L}(C_d(\mathcal{FAN}(\mathbf{F}_i^{\mathcal{D}}), d_i)]]. \qquad (5)$$

Additionally, we apply Eq. (2) to train the prediction classifier on the video-level features.

$$\mathcal{L}_{cls} = \mathbb{E}_i[\mathcal{L}(\mathcal{FAN}(\mathbf{F}_i^{\mathcal{D}}), y_i^{\mathcal{D}})]. \qquad (6)$$

To this end, we present our *MetaTrans* model.

### 3.4. Theoretical Analysis

We further provide theoretical justifications for *MetaTrans*. Firstly, we present the target error bound based on (Ben-David et al., 2006). Let $G$ be a representation map and $h \in \mathcal{H}$ be a classifier. In *MetaTrans*, it is

$$G(\mathbf{X}^{\mathcal{D}}) = \mathcal{FAN}(\mathcal{M}_1(\mathbf{X}^{\mathcal{D}} + \mathbf{P}^{\mathcal{D}}) - \mathcal{M}_2(\mathbf{X}^{\mathcal{D}})),$$

Let $\epsilon_D(h \circ G)$ denote the expected 0-1 risk on domain $\mathcal{D} \in \{\mathcal{S}, \mathcal{T}\}$. We adopt the standard $\mathcal{H}\Delta\mathcal{H}$-divergence

for domain adaptation. The following bound specializes the classical domain adaptation bound in (Ben-David et al., 2006) to our representation $G$.

**Theorem 2.** *For any feature map $G$ and any $h \in \mathcal{H}$,*

$$\epsilon_T(h \circ G) \leq \epsilon_S(h \circ G) + \frac{1}{2} d_{\mathcal{H}\Delta\mathcal{H}}(X_G^S, \ X_G^T) + \lambda^*(G),$$

*where $X_G^D$ is the marginal domain distribution with respect to $G$, and $\lambda^*(G) := \min_{h \in \mathcal{H}} (\epsilon_S(h \circ G) + \epsilon_T(h \circ G))$.*

Theorem 2 can be easily proved by applying the standard $\mathcal{H}\Delta\mathcal{H}$ adaptation bound to the induced distributions over $G$. The *MetaTrans* learning objective in Eq. (3) directly targets the first two terms in this error bound: $\mathcal{L}_{cls}$ reduces the source risk and $\mathcal{L}_{adv}$ reduces the domain discrepancy in the representation space, which theoretically ensures that *MetaTrans* provides a good feature representation for UVDA.

We then theoretically analyze why the temporal-static subtraction module in *MetaTrans* benefits the UVDA performance. We consider the factorized modeling of a video sequence where the per-frame feature follows an additive decomposition,

$$\mathbf{z}_t = \mathbf{s} + \mathbf{u}_t, \quad t = 1, \dots, T, \tag{7}$$

where $\mathbf{s} \in \mathbb{R}^d$ is a static factor (background, style, camera) and $\mathbf{u}_t$ captures the dynamics. We then propose the following theorem.

**Theorem 3.** *Let $Z_t(\mathbf{X}) := \mathcal{M}_1(\mathbf{X} + \mathbf{P})_t$ and $F_t(\mathbf{X}) = Z_t(\mathbf{X}) - \mathcal{M}_2(\mathbf{X})$. Assume that there is a static component $\mathbf{s}$ such that the static estimation error $e(\mathbf{X}) := \mathcal{M}_2(\mathbf{X}) - \mathbf{s}$ satisfies $\mathbb{E}_{\mathbf{X} \sim \mathcal{S}} \|e(\mathbf{X})\| < \infty$ and $\mathbb{E}_{\mathbf{X} \sim \mathcal{T}} \|e(\mathbf{X})\| < \infty$. Define the "ideal" residual $\tilde{F}_t(\mathbf{X}) := Z_t(\mathbf{X}) - \mathbf{s}$, so that $F_t(\mathbf{X}) = \tilde{F}_t(\mathbf{X}) - e(\mathbf{X})$. Let $P_{\mathcal{S}}^F, P_{\mathcal{T}}^F$ be the distributions of $F_t(\mathbf{X})$ under $\mathcal{S}$ and $\mathcal{T}$, and let $P_{\mathcal{S}}^{\tilde{F}}, P_{\mathcal{T}}^{\tilde{F}}$ be the corresponding distributions of $\tilde{F}_t(\mathbf{X})$. Denote the Wasserstein-1 distance as $W_1$, we then have:*

$$W_1(P_{\mathcal{S}}^F, P_{\mathcal{T}}^F) \leq W_1(P_{\mathcal{S}}^{\tilde{F}}, P_{\mathcal{T}}^{\tilde{F}}) \\ + \mathbb{E}_{\mathbf{X} \sim \mathcal{S}} \|e(\mathbf{X})\| + \mathbb{E}_{\mathbf{X} \sim \mathcal{T}} \|e(\mathbf{X})\|.$$

The proof of theorem 3 is stated in the appendix. Theorem 3 shows that the post-subtraction discrepancy between the source and target domains decomposes into (i) an *ideal temporal discrepancy* term $W_1(P_{\mathcal{S}}^{\tilde{F}}, P_{\mathcal{T}}^{\tilde{F}})$ that captures what remains after perfectly removing static nuisance, plus (ii) the static estimation errors of $\mathcal{M}_2$ on both domains. Consequently, Eq. (4) acts as a principled *preconditioning* step for domain adaptation: when $\mathcal{M}_2$ accurately estimates the static information, the additional discrepancy introduced by subtraction is small, and the remaining alignment problem is dominated by the discrepancy of the ideal dynamic residual $\tilde{F}$. This exactly aligns with our two-stream design: $\mathcal{M}_2$ removes spatial static factors, and the adversarial alignment

in Eq. (5) can focus on the residual temporal content propagated to $\mathcal{FAN}(\cdot)$, thereby reducing the discrepancy of the video-level representation used by the classifier.

**Theorem 4.** *Let a sequence $\mathbf{X}$ follow the factorized modeling $\mathbf{x}_t = \mathbf{s} + \mathbf{u}_t$, $t = 1, \dots, T$. Conditioned on the static factor $\mathbf{s}$, the dynamic factors $\{u_t\}_{t=1}^T$ satisfy $\mathbb{E}[u_t | s] = 0$, and have a finite second moment. Let $\mathcal{M}_2$ be the static stream of MetaTrans, which is temporal permutation invariant. Further define $\bar{\mathbf{u}} := \frac{1}{T} \sum_{t=1}^T \mathbf{u}_t$ and assume that each coordinate $u_{t,j}$ $(j = 1, \dots, d)$ is sub-Gaussian with parameter $\sigma^2$ conditioned on s, i.e.,*

$$\mathbb{E}\big[ \exp(\lambda u_{t,j}) \mid \mathbf{s} \big] \leq \exp\left( \frac{\sigma^2 \lambda^2}{2} \right), \quad \forall \lambda \in \mathbb{R}.$$

*Then for any $\delta \in (0, 1)$, with probability at least $1 - \delta$,*

$$\|\mathcal{M}_2(\mathbf{X}) - s\|_2 \leq \varepsilon_{\text{cal}} + L\sigma \sqrt{\frac{2d \log(2d/\delta)}{T}},$$

*where $\varepsilon_{\text{cal}}$ is a small error value and $L$ is a constant.*

The proof of theorem 4 is stated in the appendix. Theorem 4 shows that the temporal permutation-invariant $\mathcal{M}_2$ yields a reliable estimator of static factors, with the estimation error decaying as $\mathcal{O}(\sqrt{1/T})$. Combined with Theorem 3, they provide a unified mechanistic justification for *MetaTrans*: the permutation-invariant design of $\mathcal{M}_2$ yields a good modeling of static factors, which tightens the post-subtraction domain gap bound and facilitates residual representations for downstream alignment and temporal reasoning.

## 4. Experimental Study

### 4.1. Experimental Configurations

**Datasets.** Experiments are carried out on two popular UVDA benchmark datasets, namely the UCF-HMDB dataset (Chen et al., 2019) and the Epic-Kitchens dataset (Damen et al., 2018). The UCF-HMDB dataset consists of two UVDA tasks: $\mathbf{U} \rightarrow \mathbf{H}$ and $\mathbf{H} \rightarrow \mathbf{U}$. The video samples are collected by amalgamating relevant and overlapping action classes from the UCF$_{101}$ dataset (Soomro et al., 2012) and the HMDB$_{51}$ dataset (Kuehne et al., 2011). The original Epic-Kitchens dataset is a rigorous egocentric dataset featuring videos that document daily activities within kitchens. To fit the context of UVDA, (Munro & Damen, 2020) establishes three domains across the eight largest actions, **P08**, **P01**, and **P22** kitchens within the complete dataset. This results in 6 cross-domain tasks.

**Implementation Details.** Following (Wei et al., 2023), we use the I3D (Carreira & Zisserman, 2017) backbone to extract RGB features. For a given video sequence, we sample 16 frames within clips by sliding a temporal window with a temporal stride of 1. Subsequently, we feed these sliding windows to the I3D model to extract features, yielding a 2,048-dimensional feature vector for each frame of

*Table 1.* Comparison results on UCF-HMDB.

| Method & Year | Backbone | U → H | H → U | Average ↑ |
|---|---|---|---|---|
| DANN (JMLR'16) | ResNet-101 | 75.3 | 76.4 | 75.8 |
| JAN (ICML'17) | ResNet-101 | 74.7 | 76.7 | 75.7 |
| AdaBN (PR'18) | ResNet-101 | 72.2 | 77.4 | 74.8 |
| MCD (CVPR'18) | ResNet-101 | 73.9 | 79.3 | 76.6 |
| ABG (MM'20) | ResNet-101 | 79.2 | 85.1 | 82.1 |
| TCoN (AAAI'20) | ResNet-101 | 87.2 | 89.1 | 88.2 |
| MA$^2$L-TD (WACV'22) | ResNet-101 | 85.0 | 86.6 | 85.8 |
| Source-only ($\mathcal{S}_{only}$) | I3D | 80.3 | 88.8 | 84.5 |
| DANN (JMLR'16) | I3D | 80.8 | 88.1 | 84.5 |
| ADDA (CVPR'17) | I3D | 79.2 | 88.4 | 83.8 |
| TA$^3$N (ICCV'19) | I3D | 81.4 | 90.5 | 86.0 |
| SAVA (ECCV'20) | I3D | 82.2 | 91.2 | 86.7 |
| MM-SADA (CVPR' 20) | I3D | 84.2 | 91.1 | 87.7 |
| STCDA (CVPR' 21) | I3D | 83.1 | 92.1 | 87.6 |
| CoMix (NeurIPS'21) | I3D | 86.7 | 93.9 | 90.2 |
| CO$^2$A (WACV'22) | I3D | 87.8 | 95.8 | 91.8 |
| MixDANN (MM' 22) | I3D | 77.5 | 86.5 | 82.0 |
| STHC (CVPR' 23) | I3D | 90.9 | 92.1 | 91.5 |
| DALLA-V (ICCV' 23) | I3D | 88.9 | 93.1 | 91.0 |
| DFRA (JNC' 23) | I3D | 88.6 | 96.7 | 92.6 |
| TranSVAE (NeurIPs' 23) | I3D | 87.8 | **99.0** | 93.4 |
| EXTERN (TMLR' 24) | I3D | 88.9 | 91.9 | 90.4 |
| UNITE (CVPR' 24) | I3D | **92.5** | 95.0 | *93.8* |
| TAViT (DAS' 25) | I3D | 90.7 | 95.4 | 93.0 |
| Co-STAR (Arxiv' 25) | I3D | *92.4* | 94.1 | 92.6 |
| CleanAdapt (PR' 25) | I3D | 88.6 | 96.7 | 92.6 |
| MetaTrans (Ours) | I3D | 92.2 | **99.0** | **95.4** |
| Supervised-target ($\mathcal{T}_{sup}$) | I3D | 95.0 | 96.9 | 95.9 |

the video. For *MetaTrans*, the self-attention blocks consist of 4 layers, and each layer is with 8-heads attention. The standard positional embeddings stated in (Vaswani et al., 2017) are used, while other advanced positional embeddings can be easily applied. For the frame aggregation network, we use the one provided by (Chen et al., 2019).

*MetaTrans* is implemented using PyTorch. We use Adam optimizer with the weight decay of $1e^{-4}$, and set the learning rate and batch size to $1e^{-4}$ and 256, respectively. The number of epochs is set to 500 for all experiments. Target pseudo labels are incorporated starting from epoch 100. For $\lambda_1$, we adhere to the common protocol in UVDA (Sahoo et al., 2021; da Costa et al., 2022a; Wei et al., 2023), performing a grid search on the validation set. All experiments are conducted with one NVIDIA A100 GPU.

**Competitors.** We compared with multiple baselines. Initially, we consider *source-only* ($\mathcal{S}_{only}$) and *supervised-target* ($\mathcal{T}_{sup}$) baselines, which exclusively utilize labeled source data and labeled target data, respectively. These two baselines serve as the lower and upper bounds for UVDA tasks. We also include 5 popular image-based UDA methods, where temporal information is simply ignored. They are DANN (Ganin et al., 2016), JAN (Long et al., 2017), ADDA (Tzeng et al., 2017), AdaBN (Li et al., 2018), and MCD (Saito et al., 2018). Lastly and most importantly, we compare several UVDA methods, including TA$^3$N (Chen et al., 2019), SAVA (Choi et al., 2020), TCoN (Pan et al., 2020), ABG (Luo et al., 2020), MM-SADA (Munro & Damen,

2020), CoMix (Sahoo et al., 2021), STCDA (Song et al., 2021), CO$^2$A (da Costa et al., 2022a), and MA$^2$L-TD (Chen et al., 2022), MixDANN (Yin et al., 2022), DFRA (Hu & Zhu, 2023), STDN (Lin et al., 2023), TranSVAE (Wei et al., 2023), DALLA-V (Zara et al., 2023a), STHC (Li et al., 2023), UNITE (Reddy et al., 2024), EXTERN (Xu et al., 2024), HCT (Lin et al., 2024), Co-STAR (Dadashzadeh et al., 2025), C-RNA (Planamente et al., 2024), MCT (Rothenberger & Diochnos, 2025) CleanAdapt (Dasgupta et al., 2025) and TAViT (Yosry et al., 2025). For all baselines, we use RGB features as the input modality. Whenever possible, we directly compare the results by quoting the numbers reported in the corresponding papers.

**Evaluation Metrics.** We use prediction accuracy as the main metric for adaptation performance comparisons. In addition, we assess various UVDA methods while considering the number of training runs required to achieve optimal prediction accuracy. As highlighted in Introduction, the number of training runs is closely related to the number of losses in the learning objective of a method. Considering the indispensability of the task supervision loss in UVDA methods, we assign a constant weight of 1 to this term, while allowing tunable weights for other losses within the objective function across all methods. We employ identical weight for losses that serve similar learning objectives, such as adversarial loss on different feature levels. Moreover, for fair and convenient comparison, we uniformly set the number of weight value candidates to 10 for all losses in different UVDA methods. Consequently, given a UVDA method, we define the metric, *R*elative *G*ain per *R*unning *A*ttempt (*RGRA*), which accounts for both the adaptation performance gain and the number of training runs required:

$$RGRA = \frac{A_{opt} - A_{\mathcal{S}_{only}}}{A_{\mathcal{T}_{sup}} - A_{\mathcal{S}_{only}}} * \frac{1}{10 \times (N_{loss} - 1)}, \quad (8)$$

where, for a given UVDA task, $A_{opt}$ is the optimal accuracy attained by the method, $A_{\mathcal{S}only}$ is the result of $\mathcal{S}_{only}$, $A_{\mathcal{T}_{sup}}$ is for the result of $\mathcal{T}_{sup}$, and $N_{loss}$ is the number of losses employed in the method. Here, we adopt the strategy where we search for one weight while fixing the others, as opposed to the greedy search strategy. Using greedy search results in a more expensive search, where the training runs grow exponentially with $N_{loss}$.

### 4.2. Experimental Results

**Adaptation Performance Comparison.** We first compare the absolute adaptation performance among different baselines. Table 1 shows the comparison results on the UCF-HMDB dataset. The best result among all the baselines is highlighted in bold and the runner-up is highlighted in bold and italic. As can be seen, *MetaTrans* outperforms all previous methods in terms of average adaptation performance. In particular, *MetaTrans* achieves average accuracy 95.4%, improving the best competitor *UNITE* by 1.6%. Note that

*Table 2.* Comparison results on Epic-Kitchens.

| Method & Year | Backbone | P08 $\to$ P01 | P08 $\to$ P22 | P01 $\to$ P08 | P01 $\to$ P22 | P22 $\to$ P08 | P22 $\to$ P01 | Average ↑ |
|---|---|---|---|---|---|---|---|---|
| Source-only ($\mathcal{S}_{only}$) | I3D | 32.8 | 34.1 | 35.4 | 39.1 | 34.6 | 35.8 | 35.3 |
| DANN (JMLR'16) | I3D | 37.7 | 36.6 | 38.3 | 41.9 | 38.8 | 42.1 | 39.2 |
| ADDA (CVPR'17) | I3D | 35.4 | 34.9 | 36.3 | 40.8 | 36.1 | 41.4 | 37.4 |
| TA$^3$N (ICCV'19) | I3D | 34.2 | 37.4 | 40.9 | 42.8 | 39.9 | 44.2 | 39.9 |
| MM-SADA (CVPR' 20) | I3D | 45.0 | 39.7 | 41.7 | 46.1 | 42.1 | 46.1 | 43.9 |
| STCDA (CVPR' 21) | I3D | 47.7 | 41.2 | 44.4 | 47.6 | 41.1 | 45.5 | 44.6 |
| CoMix (NeurIPS'21) | I3D | 42.9 | 40.9 | 38.6 | 45.2 | 42.3 | 49.2 | 43.2 |
| MixDANN (MM' 22) | I3D | ***48.0*** | 43.0 | 43.4 | 43.9 | 43.0 | 46.9 | 44.7 |
| DFRA (JNC' 23) | I3D | 46.0 | 43.8 | 48.0 | 48.7 | 43.7 | 49.2 | 46.6 |
| TranSVAE (NeurIPs' 23) | I3D | **50.5** | 50.3 | **50.3** | **58.6** | 48.0 | **58.0** | **52.6** |
| STDN (NeurIPs' 23) | I3D | 40.5 | 38.6 | 38.5 | 44.0 | 40.4 | 47.2 | 41.6 |
| C-RNA (IJCV' 24) | I3D | 43.1 | 48.9 | 43.1 | 41.7 | **49.6** | 41.9 | 44.7 |
| HCT (TPAMI' 24) | I3D | 45.0 | 43.3 | 43.2 | 47.6 | 46.2 | 46.4 | 45.3 |
| CleanAdapt (PR' 25) | I3D | 44.5 | 40.9 | 44.6 | 45.7 | 40.7 | 47.1 | 43.9 |
| MCT (JMM' 25) | I3D | 43.2 | **51.4** | ***48.3*** | 40.8 | 48.3 | 44.6 | 46.1 |
| MetaTrans (Ours) | I3D | ***48.0*** | *50.4* | 47.4 | *56.6* | 48.5 | 55.1 | *51.0* |
| Supervised-target ($\mathcal{T}_{sup}$) | I3D | 64.0 | 63.7 | 57.0 | 63.7 | 57.0 | 64.0 | 61.5 |

*MetaTrans* achieves an improvement $2.0\%$ over *TranSVAE* that also handles the spatial and temporal divergence separately. This shows the superiority of *MetaTrans* on the UCF-HMDB benchmark, which attains the best adaptation performance with a simple learning objective.

Table 2 shows the results on the Epic-Kitchens benchmark. It can be seen that *MetaTrans* is the runner-up among all the methods on the average performance. It performs $1.6\%$ worse than TranSVAE. However, given that TranSVAE relies on seven loss terms whereas *MetaTrans* uses only two, *MetaTrans* remains highly competitive. All the comparison results in Tables 1 and 2 show that *MetaTrans* can achieve considerably good absolute adaptation performance, although it only uses the two base losses of Eqs. (1) and (2).

**Relative Gain per Running Attempts.** *RGRA* serves as a metric to evaluate the holistic performance of a UVDA method, considering both adaptation performance and the number of training runs necessary to achieve that performance. Thus, we compare *RGRA* using Eq. (8) across different baselines. The results on both UCF-HMDB and Epic-Kitchens datasets are shown in Table 3. From Table 3, it is evident that *MetaTrans* consistently outperforms all baselines, emerging as the winner across the board. Particularly noteworthy is its significant superiority over all other methods in every adaptation task. On average, *MetaTrans* can achieve adaptation performance improvements of 6.02% and 10.35% in each run attempt. Moreover, we observe that, regarding *RGRA*, the recent *TranSVAE* and *HCT* are not the state-of-the-art any more due to their complex model design, and in contrast, earlier methods with fewer losses, for example STCDA, show better *RGRA*.

**Ablation Study.** *MetaTrans* effectively addresses spatial and temporal domain disparities separately. The former is managed through the subtraction of temporal and static feature representations, while the latter is achieved via adversarial loss. In this section, we validate the efficacy of each operation through ablation studies. We propose two *MetaTrans* variants: *MetaTrans_wo_sub*, which solely focuses on temporal alignment using adversarial loss without the subtraction operation, and *MetaTrans_wo_adv*, which exclusively removes spatial domain divergence through the subtraction operation without employing adversarial loss. Comparison results of these variants with *MetaTrans* on the UCF-HMDB dataset are presented in Table 4, with the $\mathcal{S}_{only}$ included for reference. Both variants exhibit improvements over $\mathcal{S}_{only}$, highlighting the effectiveness of individual spatial and temporal divergence removal within *MetaTrans*. Furthermore, the integration of both, as in *MetaTrans*, significantly enhances adaptation performance.

**Permutation Invariance Analysis.** The key of *MetaTrans* is its temporal permutation-invariant nature. Herein, we explore two alternative model architectures for extracting static and temporal feature representations, and compare *MetaTrans* with them. The first alternative, a variant of *MetaTrans* named *MetaTrans_fs_pooling*, retains only $\mathcal{M}_1$ and adds an average pooling operation at the stream's end to derive the static feature representation. The second alternative involves utilizing a 'Bi-LSTM' to replace the transformer structure in $\mathcal{M}_1$, followed by a pooling operation for static feature extraction, referred to as *Bi-LSTM_pooling*. Neither of these alternatives is permutation-invariant. However, we apply the same adaptation idea as *MetaTrans* to address spatial and temporal divergence. We test on the UCF-HMDB dataset and show the comparison results in Table 5. As can be seen, *MetaTrans* consistently achieves notably higher accuracy than both alternatives. This superiority is primarily attributed to the permutation-invariant nature of *MetaTrans*, which implicitly enforces the 'static consistency', leading to superior static feature representation.

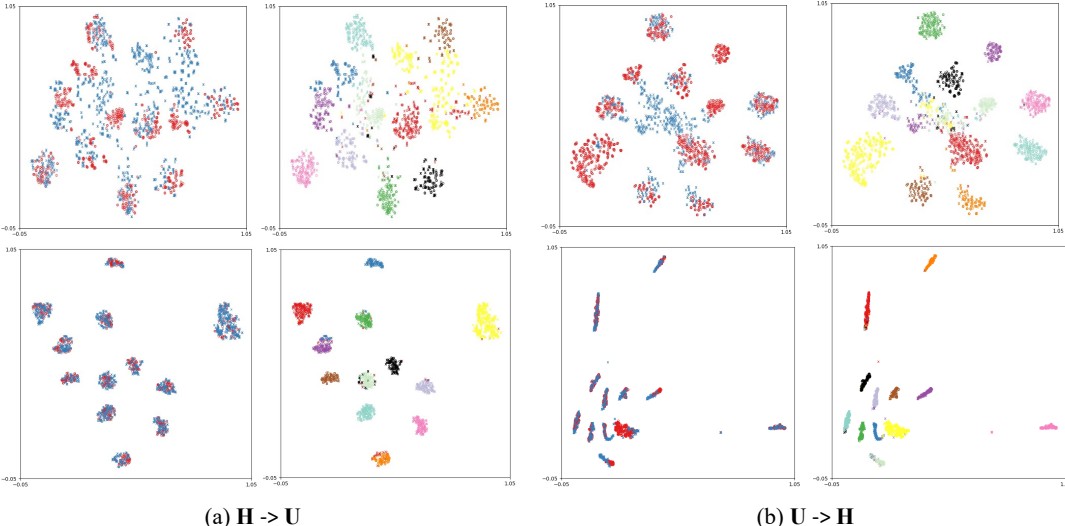

|                          |                          |
| :----------------------: | :----------------------: |
| (a) **H -> U**           | (b) **U -> H**           |

*Figure 2.* The t-SNE plots for class-wise (multi-color, the $2^{nd}$ and $4^{th}$ columns) and domain (red source & blue target, the $1^{st}$ and $3^{rd}$ columns) features for $\mathcal{S}_{only}$ (the $1^{st}$ row) and *MetaTrans* (the $2^{nd}$ row).

*Table 3. RGRA* (as in Eq. (8)) comparison results (%) on UCF-HMDB and Epic-Kitchens.

| Method | $N\downarrow$ | P08 $\rightarrow$ P01 | P08 $\rightarrow$ P22 | P01 $\rightarrow$ P08 | P01 $\rightarrow$ P22 | P22 $\rightarrow$ P08 | P22 $\rightarrow$ P01 | Average↑ | U $\rightarrow$ H | H $\rightarrow$ U | Average↑ |
| :---: | :---: | :---: | :---: | :---: | :---: | :---: | :---: | :---: | :---: | :---: | :---: |
| DANN | 2 | 1.57 | 0.84 | 1.34 | 1.14 | 1.88 | 2.23 | 1.49 | 0.38 | $-0.86$ | $-0.06$ |
| ADDA | 2 | 0.83 | 0.27 | 0.42 | 0.69 | 0.67 | 1.99 | 0.80 | $-0.75$ | $-0.43$ | $-0.63$ |
| TA3N | 3 | 0.22 | 0.56 | 1.27 | 0.75 | 1.18 | 1.49 | 0.88 | 0.38 | 1.08 | 0.63 |
| MM-SADA | 3 | 1.96 | 0.95 | 1.46 | 1.42 | 1.67 | 1.83 | 1.64 | 1.34 | 1.44 | 1.38 |
| STCDA | 3 | 2.39 | 1.20 | 2.08 | 1.73 | 1.45 | 1.72 | 1.77 | 0.97 | 2.07 | 1.37 |
| CoMix | 3 | 1.62 | 1.15 | 0.74 | 1.24 | 1.72 | 2.38 | 1.51 | 2.17 | 3.13 | 2.50 |
| MixDANN | 3 | 2.44 | 1.50 | 1.85 | 0.98 | 1.88 | 1.97 | 1.79 | $-0.94$ | $-1.39$ | $-1.10$ |
| CLDA | 3 | 1.88 | 1.15 | 2.13 | 1.34 | 1.36 | 2.00 | 1.64 | 2.83 | 4.86 | 3.57 |
| DFRA | 4 | 1.41 | 1.09 | 1.94 | 1.30 | 1.35 | 1.58 | 1.44 | 1.89 | 3.24 | 2.38 |
| TranSVAE | 5 | 1.42 | 1.37 | 1.72 | 1.98 | 1.50 | 1.97 | 1.65 | 1.27 | 3.13 | 1.94 |
| HCT | 5 | 0.98 | 0.78 | 0.90 | 0.86 | 1.29 | 0.94 | 0.95 | 2.26 | 2.10 | 2.21 |
| MetaTrans | 2 | **4.87** | **5.51** | **5.56** | **7.11** | **6.21** | **6.84** | **6.02** | **8.11** | **12.59** | **10.35** |

*Table 4.* Ablation study.

| Model Variants | Backbone | U $\rightarrow$ H | H $\rightarrow$ U | Average↑ |
| :---: | :---: | :---: | :---: | :---: |
| Source-only ($\mathcal{S}_{only}$) | I3D | 80.3 | 88.8 | 84.5 |
| MetaTrans_wo_sub | I3D | 84.5 | 92.8 | 88.7 |
| MetaTrans_wo_adv | I3D | 86.3 | 95.1 | 90.7 |
| MetaTrans | I3D | **92.2** | **99.0** | **95.4** |

*Table 5.* Permutation-invariant analysis.

| Model | Backbone | U $\rightarrow$ H | H $\rightarrow$ U | Average ↑ |
| :---: | :---: | :---: | :---: | :---: |
| Source-only ($\mathcal{S}_{only}$) | I3D | 80.3 | 88.8 | 84.5 |
| Bi-LSTM_$pooling$ | I3D | 85.8 | 94.6 | 90.2 |
| MetaTrans_$fs\_pooling$ | I3D | 84.7 | 93.9 | 89.3 |
| MetaTrans | I3D | **92.2** | **99.0** | **95.4** |

**Feature Visualization.** We further take advantage of the t-SNE (Van der Maaten & Hinton, 2008) to visualize the final video-level features learned by *MetaTrans*. Additionally, we include the t-SNE feature visualization of $\mathcal{S}_{only}$ for comparison purposes. Two sets of t-SNE figures, one using the class-wise label and another using the domain label, are shown in Figure 2. As can be seen from the t-SNE feature visualizations, *MetaTrans* produces denser and more dispersed class clusters than the source-only baseline. Within each class cluster, the source and target data are closely grouped and challenging to distinguish. All these observations well interpret the positive adaptation performance gain of *MetaTrans* over $\mathcal{S}_{only}$.

## 5. Conclusion

In this paper, we propose a frustratingly easy UVDA framework, *MetaTrans* that uses two basic losses but embodies novel UVDA idea through subtle model architecture design. Precisely, we develop a permutation-invariant temporal-static subtraction module consisting of two streams, one for static estimation and another for temporal learning. The spatial domain divergence is reduced by subtracting static features from temporal ones. Temporal alignment is then achieved by minimizing adversarial loss. Both theoretical and empirical analyses clearly verify the effectiveness of *MetaTrans* on achieving positive adaptation performance.

## Impact Statement

This paper provides a streamlined novel adaptation method for unsupervised video domain adaptation problem. It can use cross-domain video data, which effectively helps reduce the annotation efforts in related video applications. Moreover, the streamlined design of the model enables it to achieve outstanding ROI value, supporting its practical utility in real-world applications. Although the main empirical evaluation is on the video action recognition task, the model architecture and the concept proposed in this paper are also applicable to other video-related tasks, such as video semantic segmentation. More generally, the idea of removing the global information from a sequence using a transformer architecture sheds light on other related research, *e.g.*, disentanglement. The negative impacts of this work are difficult to predict. However, as a deep model, our method shares some common pitfalls of the standard deep learning models, *e.g.*, demand for powerful computing resources and vulnerability to adversarial attacks.

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

# Appendix

In this appendix, we supplement materials to support the findings and conclusions drawn in the main body of this paper. Specifically, we show the proof of theorems, and additional experimental results, followed by more discussion on *MetaTrans*.

## A. Proof of Theorems

### A.1. Proof of Temporal Permutation Invariant of $\mathcal{M}_2$

**Theorem A.1.** *Consider $\mathcal{M}_2$ composed of: (i) multi-head self-attention without positional embeddings, (ii) residual connections, (iii) layer normalization performed per feature dimension, (iv) a feature-wise feed-forward network with shared parameters across time, and (v) an average operator. Specifically, letting $\mathbf{X}$ be the input sequence and $\mathcal{T}_\pi$ be any temporal permutation, we have: $\mathcal{M}_2(\mathcal{T}_\pi(\mathbf{X})) = \mathcal{M}_2(\mathbf{X})$.*

To prove Theorem A.1, we need to prove several lemmas. We first introduce the definition of temporal permutation equivariant.

**Definition A.1.** Given a video sequence $\mathbf{X} \in \mathbb{R}^{d \times T}$ where $T$ is the temporal dimension and $d$ is the number of features and denoted $\pi$ as a permutation of $T$ elements, we define a transformation $\mathcal{T}_\pi : \mathbb{R}^{d \times T} \to \mathbb{R}^{d \times T}$ a temporal permutation if $\mathcal{T}_\pi(\mathbf{X}) = \mathbf{X}\mathbf{P}_\pi$ where $\mathbf{P}_\pi \in \mathbb{R}^{T \times T}$ denotes the permutation matrix associated with $\pi$ defined as $\mathbf{P}_\pi = [\mathbf{e}_{\pi(1)}, \mathbf{e}_{\pi(2)}, ..., \mathbf{e}_{\pi(T)}]$ and $\mathbf{e}_{\pi(i)}$ is a one-hot vector of length $T$ with its $i$-*th* element being 1. An operator $\mathcal{O} : \mathbb{R}^{d \times T} \to \mathbb{R}^{d \times T}$ is temporally permutation equivariant if $\mathcal{O}(\mathcal{T}_\pi(\mathbf{X})) = \mathcal{T}_\pi(\mathcal{O}(\mathbf{X}))$ for any $\mathbf{X}$ and any temporal permutation $\mathcal{T}_\pi$.

**Lemma A.1.** *A self-attention operator $\mathcal{SA}(\cdot)$ is permutation equivariant. Specifically, letting $\mathbf{X}$ denote the input matrix and $\mathcal{T}_\pi$ denotes any temporal permutation, we have: $\mathcal{SA}(\mathcal{T}_\pi(\mathbf{X})) = \mathcal{T}_\pi(\mathcal{SA}(\mathbf{X}))$.*

*Proof.* When applying a temporal permutation $\mathcal{T}_\pi$ to the input sequence $\mathbf{X}$ of a self-attention operator $\mathcal{SA}$, we have[1]:

$$\mathcal{SA}(\mathcal{T}_\pi(\mathbf{X})) = \mathbf{W}_v \mathcal{T}_\pi(\mathbf{X}) \cdot \text{softmax}\left((\mathbf{W}_k \mathcal{T}_\pi(\mathbf{X}))^\top \cdot \mathbf{W}_q \mathcal{T}_\pi(\mathbf{X})\right)$$

$$= \mathbf{W}_v \mathbf{X}\mathbf{P}_\pi \cdot \text{softmax}\left((\mathbf{W}_k \mathbf{X}\mathbf{P}_\pi)^\top \cdot \mathbf{W}_q \mathbf{X}\mathbf{P}_\pi\right)$$

$$= \mathbf{W}_v \mathbf{X}\mathbf{P}_\pi \cdot \text{softmax}\left(\mathbf{P}_\pi^\top (\mathbf{W}_k \mathbf{X})^\top \cdot \mathbf{W}_q \mathbf{X}\mathbf{P}_\pi\right).$$

It is easy to verify that:

$$\text{softmax}\left(\mathbf{P}_\pi^\top \mathbf{M}\mathbf{P}_\pi\right) = \mathbf{P}_\pi^\top \text{softmax}\left(\mathbf{M}\right)\mathbf{P}_\pi,$$

since $\mathbf{P}_\pi$ is an orthogonal matrix and $\mathbf{P}_\pi \mathbf{P}_\pi^\top = \mathbf{I}$. Thus, we have:

$$\mathcal{SA}(\mathcal{T}_\pi(\mathbf{X})) = \mathbf{W}_v \mathbf{X}\mathbf{P}_\pi \cdot \text{softmax}\left(\mathbf{P}_\pi^\top (\mathbf{W}_k \mathbf{X})^\top \cdot \mathbf{W}_q \mathbf{X}\mathbf{P}_\pi\right)$$

$$= \mathbf{W}_v \mathbf{X}\mathbf{P}_\pi \mathbf{P}_\pi^\top \cdot \text{softmax}\left((\mathbf{W}_k \mathbf{X})^\top \cdot \mathbf{W}_q \mathbf{X}\right)\mathbf{P}_\pi$$

$$= \mathbf{W}_v \mathbf{X} \cdot \text{softmax}\left((\mathbf{W}_k \mathbf{X})^\top \cdot \mathbf{W}_q \mathbf{X}\right)\mathbf{P}_\pi$$

$$= \mathcal{T}_\pi(\mathcal{SA}(\mathbf{X})).$$

To this end, we complete the proof. □

**Lemma A.2.** *Let $\mathbf{P}_\pi \in \mathbb{R}^{T \times T}$ be a permutation matrix. Let $\mathbf{X}^1 \in \mathbb{R}^{T \times d_1}, \ldots, \mathbf{X}^H \in \mathbb{R}^{T \times d_H}$ be matrices with the same number of rows. Define feature-wise concatenation: $\text{Concat}(\mathbf{X}^1, \ldots, \mathbf{X}^H) \in \mathbb{R}^{T \times (d_1 + \cdots + d_H)}$ by concatenating columns. Then*

$$\mathbf{P}_\pi \text{Concat}(\mathbf{X}^1, \ldots, \mathbf{X}^H) = \text{Concat}(\mathbf{P}_\pi \mathbf{X}^1, \ldots, \mathbf{P}_\pi \mathbf{X}^H). \tag{1}$$

*Proof.* Let $\mathbf{C} = \text{Concat}(\mathbf{X}^1, \ldots, \mathbf{X}^H)$ and $\mathbf{C}' = \text{Concat}(\mathbf{P}_\pi \mathbf{X}^1, \ldots, \mathbf{P}_\pi \mathbf{X}^H)$. We show that equality holds row by row.

---

[1] $\mathbf{W}_q, \mathbf{W}_k, \mathbf{W}_v$ are the query, key, value parameter matrices, respectively. Please refer (Vaswani et al., 2017) for more details.

Let $\pi : \{1, \ldots, T\} \to \{1, \ldots, T\}$ be the permutation represented by $P_\pi$, defined so that the $t$-th column of $\mathbf{P}_\pi \mathbf{X}$ is the $\pi(t)$-th column of $\mathbf{X}$:

$$(\mathbf{P}_\pi \mathbf{X})_{t,:} = \mathbf{X}_{\pi(t),:}, \qquad \forall \mathbf{X} \in \mathbb{R}^{T \times d}.$$

Then the $t$-th row of the left-hand side of Eq. (1) is

$$(\mathbf{P}_\pi \mathbf{C})_{t,:} = \mathbf{C}_{\pi(t),:} = \left[ \mathbf{X}^1_{\pi(t),:} \quad \cdots \quad \mathbf{X}^H_{\pi(t),:} \right], \tag{2}$$

where $\left[ \cdot \; \cdots \; \cdot \right]$ denotes the concatenation of row vectors.

On the other hand, the $t$-th row of the right-hand side of Eq. (1) is

$$\mathbf{C}'_{t,:} = \left[ (\mathbf{P}_\pi \mathbf{X}^1)_{t,:} \quad \cdots \quad (\mathbf{P}_\pi \mathbf{X}^H)_{t,:} \right] = \left[ \mathbf{X}^1_{\pi(t),:} \quad \cdots \quad \mathbf{X}^H_{\pi(t),:} \right]. \tag{3}$$

Comparing Eq. (2) and Eq. (3) shows $(\mathbf{P}_\pi \mathbf{C})_{t,:} = \mathbf{C}'_{t,:}$ for all $t$, hence proving Eq. (1). $\qquad\square$

**Lemma A.3.** *Let $\mathcal{MHA}(X)$ denote standard multi-head self-attention:*

$$\mathcal{MHA}(X) = \mathrm{Concat}(\mathcal{SA}_1(\mathbf{X}), \ldots, \mathcal{SA}_H(\mathbf{X}))\mathbf{W}_O,$$

*where each $\mathcal{SA}_h$ has the form in Lemma A.1 and $\mathbf{W}_O$ is the output projection. Then for any permutation $\mathcal{T}_\pi$ with the permutation matrix $\mathbf{P}_\pi$,*

$$\mathcal{MHA}(\mathcal{T}_\pi(\mathbf{X})) = \mathcal{T}_\pi(\mathcal{MHA}(\mathbf{X})).$$

*Proof.* By Lemma A.1, $\mathcal{SA}_h(\mathcal{T}_\pi(\mathbf{X})) = \mathcal{T}_\pi(\mathcal{SA}_h(\mathbf{X}))$ for every head. According to Lemma A.2,

$$\mathrm{Concat}(\mathcal{SA}_1(\mathcal{T}_\pi(\mathbf{X})), \ldots, \mathcal{SA}_H(\mathcal{T}_\pi(\mathbf{X}))) = \mathcal{T}_\pi(\mathrm{Concat}(\mathcal{SA}_1(\mathbf{X}), \ldots, \mathcal{SA}_H(\mathbf{X}))).$$

Right-multiplication by $\mathbf{W}_O$ acts identically on each row, hence preserving equivariance. To this end, we complete the proof. $\qquad\square$

**Lemma A.4.** *Let $\phi : \mathbb{R}^d \to \mathbb{R}^{d'}$ be any function and define the feature-wise extension $\Phi : \mathbb{R}^{T \times d} \to \mathbb{R}^{T \times d'}$ by $(\Phi(\mathbf{X}))_{t,:} = \phi(\mathbf{X}_{t,:})$ for all $t$. Then for any permutation matrix $\mathbf{P}_\pi$,*

$$\Phi(\mathbf{P}_\pi \mathbf{X}) = \mathbf{P}_\pi \, \Phi(\mathbf{X}).$$

*Proof.* Let $\mathbf{X}' = \mathbf{P}_\pi \mathbf{X}$. The $t$-th row of $\mathbf{X}'$ equals the $\pi(t)$-th row of $\mathbf{X}$. Hence

$$(\Phi(\mathbf{X}'))_{t,:} = \phi(\mathbf{X}'_{t,:}) = \phi(\mathbf{X}_{\pi(t),:}) = (\Phi(\mathbf{X}))_{\pi(t),:} = (\mathbf{P}_\pi \Phi(\mathbf{X}))_{t,:}.$$

Thus $\Phi(\mathbf{P}_\pi \mathbf{X}) = \mathbf{P}_\pi \Phi(\mathbf{X})$. $\qquad\square$

Lemma A.4 applies directly to the standard FFN and to feature-wise LayerNorm. Residual addition also preserves equivariance because $(\mathbf{P}_\pi A) + (\mathbf{P}_\pi B) = \mathbf{P}_\pi (A + B)$.

**Lemma A.5.** *Consider a Transformer encoder block*

$$\mathcal{B}(\mathbf{X}) = \mathcal{LN}\big(\mathbf{X} + \mathcal{MHA}(\mathcal{LN}(\mathbf{X}))\big) \quad \text{followed by} \quad \mathcal{LN}\big(\cdot + \mathcal{FFN}(\mathcal{LN}(\cdot))\big),$$

*where $\mathcal{MHA}$ is multi-head self-attention without positional embeddings, $\mathcal{FFN}$ is a feature-wise feed-forward network, and $\mathcal{LN}$ is feature-wise layer normalization. Then for any permutation matrix $\mathbf{P}_\pi$,*

$$\mathcal{B}(\mathbf{P}_\pi \mathbf{X}) = \mathbf{P}_\pi \mathcal{B}(\mathbf{X}). \tag{4}$$

*Moreover, any stack $\mathcal{T} = \mathcal{B}_L \circ \cdots \circ \mathcal{B}_1$ is also permutation equivariant: $\mathcal{T}(\mathbf{P}_\pi \mathbf{X}) = \mathbf{P}_\pi \mathcal{T}(\mathbf{X})$.*

*Proof.* As the stacked case follows by induction using closure under composition, we prove equivariance for one block. Let $\mathbf{X}' = \mathbf{P}_\pi \mathbf{X}$. First, feature-wise LayerNorm is permutation equivariant by Lemma A.4 (with $\phi$ being $\mathcal{LN}$), so $\mathcal{LN}(\mathbf{X}') = \mathbf{P}_\pi \mathcal{LN}(\mathbf{X})$. By Lemma A.3, $\mathcal{MHA}(\mathcal{LN}(\mathbf{X}')) = \mathcal{MHA}(\mathbf{P}_\pi \mathcal{LN}(\mathbf{X})) = \mathbf{P}_\pi \mathcal{MHA}(\mathcal{LN}(\mathbf{X}))$. Therefore the first residual branch is equivariant:

$$\mathbf{X}' + \mathcal{MHA}(\mathcal{LN}(\mathbf{X}')) = \mathbf{P}_\pi \mathbf{X} + \mathbf{P}_\pi \mathcal{MHA}(\mathcal{LN}(\mathbf{X})) = \mathbf{P}_\pi \big(\mathbf{X} + \mathcal{MHA}(\mathcal{LN}(\mathbf{X}))\big).$$

Applying feature-wise LayerNorm again preserves equivariance, giving

$$\mathcal{LN}\big(\mathbf{X}' + \mathcal{MHA}(\mathcal{LN}(\mathbf{X}'))\big) = \mathbf{P}_\pi \mathcal{LN}\big(\mathbf{X} + \mathcal{MHA}(\mathcal{LN}(\mathbf{X}))\big).$$

Denote this intermediate output by $\mathbf{Y}$. Then $\mathbf{Y}' = \mathbf{P}_\pi \mathbf{Y}$.

For the FFN sub-layer, $\mathcal{FFN}$ is feature-wise (same parameters applied to each row), hence permutation equivariant by Lemma A.4. Also, $\mathcal{LN}$ is temporal permutation equivariant. Thus

$$\mathcal{FFN}(\mathcal{LN}(\mathbf{Y}')) = \mathcal{FFN}(\mathbf{P}_\pi \mathcal{LN}(\mathbf{Y})) = \mathbf{P}_\pi \mathcal{FFN}(\mathcal{LN}(\mathbf{Y})).$$

The second residual branch is equivariant:

$$\mathbf{Y}' + \mathcal{FFN}(\mathcal{LN}(\mathbf{Y}')) = \mathbf{P}_\pi \mathbf{Y} + \mathbf{P}_\pi \mathcal{FFN}(\mathcal{LN}(\mathbf{Y})) = \mathbf{P}_\pi \big(\mathbf{Y} + \mathcal{FFN}(\mathcal{LN}(\mathbf{Y}))\big),$$

and applying the final feature-wise LayerNorm preserves equivariance. Therefore $\mathcal{B}(\mathbf{P}_\pi \mathbf{X}) = \mathbf{P}_\pi \mathcal{B}(\mathbf{X})$, proving Eq. (4).

For a stack $\mathcal{T} = \mathcal{B}_L \circ \cdots \circ \mathcal{B}_1$, assume $\mathcal{T}_{k-1}(\mathbf{P}_\pi \mathbf{X}) = \mathbf{P}_\pi \mathcal{T}_{k-1}(\mathbf{X})$ holds. Then

$$\mathcal{T}_k(\mathbf{P}_\pi \mathbf{X}) = \mathcal{B}_k(\mathcal{T}_{k-1}(\mathbf{P}_\pi \mathbf{X})) = \mathcal{B}_k(\mathbf{P}_\pi \mathcal{T}_{k-1}(\mathbf{X})) = \mathbf{P}_\pi \mathcal{B}_k(\mathcal{T}_{k-1}(\mathbf{X})) = \mathbf{P}_\pi \mathcal{T}_k(\mathbf{X}),$$

where we used equivariance of $\mathcal{B}_k$. This completes the proof. $\qquad\square$

**Lemma A.6.** *Let $\mathcal{O}$ be any permutation equivariant operator, and define mean pooling $\mathcal{AV}(\mathbf{X}) = \frac{1}{T}\mathbf{1}^\top \mathbf{X} \in \mathbb{R}^{1\times d}$ (averaging rows). Then the pooled representation is temporal permutation invariant: $\mathcal{AV}(\mathcal{O}(\mathbf{P}_\pi \mathbf{X})) = \mathcal{AV}(\mathcal{O}(\mathbf{X}))$ for all $\mathbf{X}, \mathbf{P}_\pi$.*

*Proof.* Since $\mathcal{O}$ is permutations equivariant, $\mathcal{O}(\mathbf{P}_\pi \mathbf{X}) = \mathbf{P}_\pi \mathcal{O}(\mathbf{X})$. Mean pooling is invariant to row permutations because $\mathbf{1}^\top \mathbf{P}_\pi = \mathbf{1}^\top$ for any permutation matrix $\mathbf{P}_\pi$. Therefore,

$$\mathcal{AV}(\mathcal{O}(\mathbf{P}_\pi \mathbf{X})) = \frac{1}{T}\mathbf{1}^\top \mathcal{O}(\mathbf{P}_\pi \mathbf{X}) = \frac{1}{T}\mathbf{1}^\top \mathbf{P}_\pi \mathcal{O}(\mathbf{X}) = \frac{1}{T}\mathbf{1}^\top \mathcal{O}(\mathbf{X}) = \mathcal{AV}(\mathcal{O}(\mathbf{X})).$$

$\square$

With Lemmas 1 to 6, we can prove Theorem A.1.

### A.2. Proof of Post-subtraction Distribution Discrepancy Bound

**Theorem A.2.** *Let $Z_t(\mathbf{X}) := \mathcal{M}_1(\mathbf{X} + \mathbf{P})_t$ and $F_t(\mathbf{X}) = Z_t(\mathbf{X}) - \mathcal{M}_2(\mathbf{X})$. Assume that there is a static component $\mathbf{s}$ such that the static estimation error $e(\mathbf{X}) := \mathcal{M}_2(\mathbf{X}) - \mathbf{s}$ satisfies $\mathbb{E}_{\mathbf{X}\sim\mathcal{S}}\|e(\mathbf{X})\| < \infty$ and $\mathbb{E}_{\mathbf{X}\sim\mathcal{T}}\|e(\mathbf{X})\| < \infty$. Define the "ideal" residual $\tilde{F}_t(\mathbf{X}) := Z_t(\mathbf{X}) - \mathbf{s}$, so that $F_t(\mathbf{X}) = \tilde{F}_t(\mathbf{X}) - e(\mathbf{X})$. Let $P_S^F, P_T^F$ be the distributions of $F_t(\mathbf{X})$ under $\mathcal{S}$ and $\mathcal{T}$, and let $P_S^{\tilde{F}}, P_T^{\tilde{F}}$ be the corresponding distributions of $\tilde{F}_t(\mathbf{X})$. Denote the Wasserstein-1 distance as $W_1$, we then have:*

$$W_1(P_S^F, P_T^F) \le W_1(P_S^{\tilde{F}}, P_T^{\tilde{F}}) + \mathbb{E}_{\mathbf{X}\sim\mathcal{D}_S}\|e(\mathbf{X})\| + \mathbb{E}_{\mathbf{X}\sim\mathcal{D}_T}\|e(\mathbf{X})\|. \tag{5}$$

*Proof.* To prove the theorem, we first present the following three facts.

*Fact 1 (Kantorovich–Rubinstein duality (Arjovsky et al., 2017)).* For probability measures $P, Q$ on $\mathbb{R}^m$ with finite first moments,

$$W_1(P, Q) = \sup_{\|f\|_{\mathrm{Lip}}\le 1} \left(\mathbb{E}_{U\sim P}[f(U)] - \mathbb{E}_{V\sim Q}[f(V)]\right), \tag{6}$$

where $\|f\|_{\mathrm{Lip}} \leq 1$ means $|f(u) - f(v)| \leq \|u - v\|$ for all $u, v$.

*Fact 2 (Triangle inequality).* $W_1$ is a metric, hence for any measures $P, Q, R$,

$$W_1(P, Q) \leq W_1(P, R) + W_1(R, Q). \tag{7}$$

*Fact 3 (Lipschitz perturbation bound).* If $f$ is 1-Lipschitz, then for any random vectors $U, V$,

$$|f(U) - f(V)| \leq \|U - V\| \quad \Rightarrow \quad |\mathbb{E}[f(U)] - \mathbb{E}[f(V)]| \leq \mathbb{E}\|U - V\|. \tag{8}$$

By definition,
$$F_t(\mathbf{X}) = Z_t(\mathbf{X}) - \mathcal{M}_2(\mathbf{X}) = Z_t(\mathbf{X}) - \mathbf{s} - (\mathcal{M}_2(\mathbf{X}) - \mathbf{s}) = \tilde{F}_t(\mathbf{X}) - e(\mathbf{X}). \tag{9}$$

Applying Fact 2 twice with intermediate measures $P_S^{\tilde{F}}$ and $P_T^{\tilde{F}}$ gives

$$W_1(P_S^F, P_T^F) \leq W_1(P_S^F, P_S^{\tilde{F}}) + W_1(P_S^{\tilde{F}}, P_T^{\tilde{F}}) + W_1(P_T^{\tilde{F}}, P_T^F). \tag{10}$$

By Fact 1,
$$W_1(P_S^F, P_S^{\tilde{F}}) = \sup_{\|f\|_{\mathrm{Lip}} \leq 1} \left( \mathbb{E}_{\mathbf{X} \sim \mathcal{D}_S}[f(F_t(\mathbf{X}))] - \mathbb{E}_{\mathbf{X} \sim \mathcal{D}_S}[f(\tilde{F}_t(\mathbf{X}))] \right).$$

Using Fact 3 and Eq. (9),

$$\begin{aligned}
\mathbb{E}_{\mathbf{X} \sim \mathcal{D}_S}[f(F_t(\mathbf{X}))] - \mathbb{E}_{\mathbf{X} \sim \mathcal{D}_S}[f(\tilde{F}_t(\mathbf{X}))] &\leq \mathbb{E}_{\mathbf{X} \sim \mathcal{D}_S} |f(F_t(\mathbf{X})) - f(\tilde{F}_t(\mathbf{X}))| \\
&\leq \mathbb{E}_{\mathbf{X} \sim \mathcal{D}_S} \|F_t(\mathbf{X}) - \tilde{F}_t(\mathbf{X})\| \\
&= \mathbb{E}_{\mathbf{X} \sim \mathcal{D}_S} \|e(\mathbf{X})\|.
\end{aligned} \tag{11}$$

Since the bound holds for every 1-Lipschitz $f$, taking the supremum yields

$$W_1(P_S^F, P_S^{\tilde{F}}) \leq \mathbb{E}_{\mathbf{X} \sim \mathcal{D}_S} \|e(\mathbf{X})\|. \tag{12}$$

Similarly, we can obtain
$$W_1(P_T^{\tilde{F}}, P_T^F) \leq \mathbb{E}_{\mathbf{X} \sim \mathcal{D}_T} \|e(\mathbf{X})\|. \tag{13}$$

Substitute Eq. (12) and Eq. (13) in Eq. (10) to obtain

$$W_1(P_S^F, P_T^F) \leq W_1(P_S^{\tilde{F}}, P_T^{\tilde{F}}) + \mathbb{E}_{\mathbf{X} \sim \mathcal{D}_S} \|e(\mathbf{X})\| + \mathbb{E}_{\mathbf{X} \sim \mathcal{D}_T} \|e(\mathbf{X})\|,$$

which completes the proof. $\qquad\square$

### A.3. Proof of Estimation Error Bound on Static Factor

**Theorem A.3.** *Let a sequence* $\mathbf{X}$ *follow the factorized modeling* $\mathbf{x}_t = \mathbf{s} + \mathbf{u}_t$, $t = 1, \ldots, T$. *Conditioned on the static factor* $\mathbf{s}$, *the dynamic factors* $\{u_t\}_{t=1}^T$ *satisfy* $\mathbb{E}[u_t|s] = 0$, *and have a finite second moment. Let* $\mathcal{M}_2$ *be the static stream of* MetaTrans, *which is temporal permutation invariant. Further define* $\bar{\mathbf{u}} := \frac{1}{T} \sum_{t=1}^T \mathbf{u}_t$ *and assume that each coordinate* $u_{t,j}$ *($j = 1, \ldots, d$) is sub-Gaussian with parameter* $\sigma^2$ *conditioned on $s$, i.e.,*

$$\mathbb{E}\big[\exp(\lambda u_{t,j}) \mid \mathbf{s}\big] \leq \exp\left(\frac{\sigma^2 \lambda^2}{2}\right), \quad \forall \lambda \in \mathbb{R}.$$

*Then for any* $\delta \in (0, 1)$, *with probability at least* $1 - \delta$ *(conditioned on* $\mathbf{s}$),

$$\|\mathcal{M}_2(\mathbf{X}) - s\|_2 \leq \varepsilon_{\mathrm{cal}} + L\sigma \sqrt{\frac{2d \log(2d/\delta)}{T}}. \tag{14}$$

*Proof.* Since $\mathcal{M}_2$ is temporal permutation invariant, it motivates the following conditions.

(C1) Approximate calibration on constant sequences: there exists $\varepsilon_{\text{cal}} \geq 0$ such that

$$\|\mathcal{M}_2(\mathbf{s}\mathbf{1}^\top) - \mathbf{s}\|_2 \leq \varepsilon_{\text{cal}}. \tag{15}$$

(C2) Mean-stability: there exists $L > 0$ such that for all $X$ and all such $s$,

$$\|\mathcal{M}_2(\mathbf{X}) - \mathcal{M}_2(\mathbf{s}\mathbf{1}^\top)\|_2 \leq L \left\| \frac{1}{T} \sum_{t=1}^{T} (\mathbf{x}_t - \mathbf{s}) \right\|_2. \tag{16}$$

We add and subtract $\mathcal{M}_2(\mathbf{s}\mathbf{1}^\top)$ and apply the triangle inequality,

$$\|\mathcal{M}_2(\mathbf{X}) - \mathbf{s}\|_2 \leq \|\mathcal{M}_2(\mathbf{X}) - M_2(\mathbf{s}\mathbf{1}^\top)\|_2 + \|\mathcal{M}_2(s\mathbf{1}^\top) - \mathbf{s}\|_2. \tag{17}$$

By (C2),

$$\|\mathcal{M}_2(\mathbf{X}) - \mathcal{M}_2(s\mathbf{1}^\top)\|_2 \leq L \left\| \frac{1}{T} \sum_{t=1}^{T} (\mathbf{x}_t - \mathbf{s}) \right\|_2 = L \left\| \frac{1}{T} \sum_{t=1}^{T} \mathbf{u}_t \right\|_2 = L\|\bar{\mathbf{u}}\|_2.$$

With (C1), substituting into Eq. (17) yields the core bound

$$\|\mathcal{M}_2(\mathbf{X}) - \mathbf{s}\|_2 \leq L\|\bar{\mathbf{u}}\|_2 + \varepsilon_{\text{cal}}. \tag{18}$$

Fix $j \in \{1, \ldots, d\}$ and define $\bar{u}_j := \frac{1}{T} \sum_{t=1}^{T} u_{t,j}$. Conditioned on $\mathbf{s}$, for any $\lambda \in \mathbb{R}$,

$$\mathbb{E}[\exp(\lambda \bar{u}_j) \mid \mathbf{s}] = \mathbb{E}\left[ \exp\left( \frac{\lambda}{T} \sum_{t=1}^{T} u_{t,j} \right) \mid \mathbf{s} \right]$$

$$= \prod_{t=1}^{T} \mathbb{E}\left[ \exp\left( \frac{\lambda}{T} u_{t,j} \right) \mid \mathbf{s} \right]$$

$$\leq \prod_{t=1}^{T} \exp\left( \frac{\sigma^2}{2} \left( \frac{\lambda}{T} \right)^2 \right) = \exp\left( \frac{\sigma^2 \lambda^2}{2T} \right). \tag{19}$$

Let $a > 0$ and $\lambda > 0$. By Markov's inequality,

$$\Pr(\bar{u}_j \geq a \mid \mathbf{s}) = \Pr\left( \exp(\lambda \bar{u}_j) \geq \exp(\lambda a) \mid \mathbf{s} \right)$$
$$\leq \exp(-\lambda a) \, \mathbb{E}[\exp(\lambda \bar{u}_j) \mid \mathbf{s}]$$
$$\leq \exp\left( -\lambda a + \frac{\sigma^2 \lambda^2}{2T} \right). \tag{20}$$

We minimize the exponent in Eq. (20) over $\lambda > 0$:

$$\psi(\lambda) := -\lambda a + \frac{\sigma^2 \lambda^2}{2T}, \qquad \psi'(\lambda) = -a + \frac{\sigma^2}{T} \lambda.$$

Setting $\psi'(\lambda) = 0$ yields $\lambda^\star = \frac{Ta}{\sigma^2}$, and substituting back gives

$$\psi(\lambda^\star) = -\frac{Ta^2}{\sigma^2} + \frac{\sigma^2}{2T} \cdot \frac{T^2 a^2}{\sigma^4} = -\frac{Ta^2}{2\sigma^2}.$$

Therefore,

$$\Pr(\bar{u}_j \geq a \mid \mathbf{s}) \leq \exp\left( -\frac{Ta^2}{2\sigma^2} \right). \tag{21}$$

Applying the same bound to $-\bar{u}_j$ gives $\Pr(\bar{u}_j \leq -a \mid \mathbf{s}) \leq \exp(-Ta^2/(2\sigma^2))$, hence

$$\Pr(|\bar{u}_j| \geq a \mid \mathbf{s}) \leq 2\exp\left(-\frac{Ta^2}{2\sigma^2}\right). \tag{22}$$

Consider the Union bound for $\|\bar{\mathbf{u}}\|_\infty$. Let $\|\bar{\mathbf{u}}\|_\infty := \max_{1 \leq j \leq d'} |\bar{u}_j|$. Then

$$\Pr(\|\bar{\mathbf{u}}\|_\infty \geq a \mid \mathbf{s}) = \Pr\left(\bigcup_{j=1}^{d'} \{|\bar{u}_j| \geq a\} \,\Big|\, \mathbf{s}\right)$$

$$\leq \sum_{j=1}^{d'} \Pr(|\bar{u}_j| \geq a \mid \mathbf{s}) \leq 2d' \exp\left(-\frac{Ta^2}{2\sigma^2}\right). \tag{23}$$

Now convert $\ell_\infty$ to $\ell_2$. For any $\mathbf{v} \in \mathbb{R}^d$, $\|\mathbf{v}\|_2 \leq \sqrt{d}\|\mathbf{v}\|_\infty$. Hence the event inclusion holds:

$$\{\|\bar{\mathbf{u}}\|_2 \geq \varepsilon\} \subseteq \left\{\|\bar{\mathbf{u}}\|_\infty \geq \varepsilon/\sqrt{d}\right\}.$$

Applying Eq. (23) with $a = \varepsilon/\sqrt{d}$ yields

$$\Pr(\|\bar{\mathbf{u}}\|_2 \geq \varepsilon \mid \mathbf{s}) \leq 2d \exp\left(-\frac{T(\varepsilon^2/d)}{2\sigma^2}\right) = 2d \exp\left(-\frac{T\varepsilon^2}{2d\sigma^2}\right). \tag{24}$$

Now choose $\varepsilon$ such that the right-hand side equals $\delta$:

$$2d \exp\left(-\frac{T\varepsilon^2}{2d\sigma^2}\right) = \delta \quad\Longleftrightarrow\quad \varepsilon = \sigma\sqrt{\frac{2d\log(2d/\delta)}{T}}.$$

Then Eq. (24) becomes

$$\Pr\left(\|\bar{\mathbf{u}}\|_2 \geq \sigma\sqrt{\frac{2d\log(2d/\delta)}{T}} \,\Big|\, \mathbf{s}\right) \leq \delta. \tag{25}$$

Finally, combining Eq. (25) with Eq. (18) proves Eq. (14). $\qquad\square$

**Why permutation invariance motivates C1 and C2.** Because $\mathcal{M}_2$ uses a Transformer encoder without positional embeddings and pools over time, it is invariant to temporal permutations: $\mathcal{M}_2(\mathbf{X}\mathbf{P}_\pi) = \mathcal{M}_2(\mathbf{X})$. This invariance places $\mathcal{M}_2$ in the correct symmetry class for estimating a static (time-order–independent) nuisance, but invariance alone does not enforce correctness or stability. We therefore impose two additional properties. C1 is a calibration condition requiring $\mathcal{M}_2$ to act as (approximate) identity on constant sequences $\mathbf{s}\mathbf{1}^\top$, corresponding to the "no-dynamics" case. C2 is a mean-stability condition stating that deviations from a constant sequence affect the output primarily through the average perturbation, which is the canonical permutation-invariant statistic that concentrates at rate $O(1/\sqrt{T})$ under i.i.d. mean-zero fluctuations. Together, permutation invariance + C1 + C2 formalize that $\mathcal{M}_2$ implements a stable static estimator.

## B. Experimental Results

### B.1. Complexity Analysis

We conduct complexity analysis on *MetaTrans*. We compare it with TA$^3$N, CO$^2$A, CoMix, and *TranSVAE* on the number of trainable parameters, multiply-accumulate operations (MACs), floating-point operations (FLOPs), and inference frame-per-second (FPS). We show comparison results in Table 1. It can be seen that *MetaTrans* requires less trainable parameters than *TranSVAE*, CO$^2$A, and CoMix. Regarding MACs and FLOPs, they are competitive among different methods due to the usage of the same I3D backbone. For UNITE, it is based on Vit-B/16 with 3 training stages using 4 GPU, and thus is more computationally expensive. In FPS evaluations, *MetaTrans* outperforms all baselines, demonstrating superior efficiency with fewer training runs and the fastest inference speeds.

*Table 1.* Model complexity analysis.

| Methods | Trainable Params | MACs | FLOPs | FPS |
|---|---|---|---|---|
| TA$^3$N | 7.69 M | 18.2318 G | 36.4636 G | 0.0134 s |
| CoMix | 30.37 M | 18.5640 G | 37.1280 G | 0.0157 s |
| CO$^2$A | 23.67 M | 18.1884 G | 36.3768 G | 0.0127 s |
| TranSVAE | 12.74 M | 18.2657 G | 36.5314 G | 0.0133 s |
| UNITE | 86.00 M | 27.7078 G | 55.4156 G | 0.0164 s |
| MetaTrans | 8.67 M | 18.2458 G | 36.4916 G | 0.0091 s |

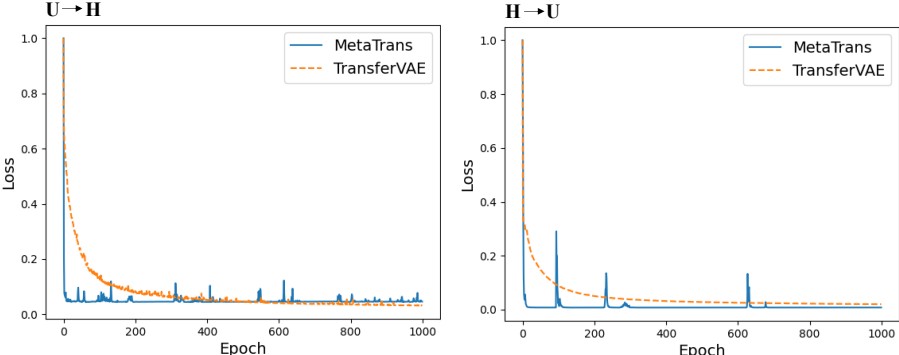

*Figure 1.* Learning curves on UCF-HMDB.

We also compare the convergence speed between *MetaTrans* and the latest *TranSVAE* on the UCF-HMDB dataset. The learning curves of the two methods are shown in Figure 1. As observed, *MetaTrans* converges more rapidly than *TranSVAE*, necessitating fewer learning epochs. This speed discrepancy can be attributed to the simpler architecture of *MetaTrans*, which involves only two primary loss terms, as opposed to *TranSVAE*'s balancing act with five terms, including MSE and KL-divergence losses. This observation underscores the training efficiency advantage of *MetaTrans*.

## B.2. Ablation Study.

We further conduct ablation study on the Epic-Kitchens dataset. Recall the two *MetaTrans* variants: *MetaTrans_wo_sub*, which solely focuses on temporal alignment using adversarial loss without the subtraction operation, and *MetaTrans_wo_adv*, which exclusively removes spatial domain divergence through the subtraction operation without employing adversarial loss. Comparison results of these variants with *MetaTrans* on Epic-Kitchens dataset are presented in Table 2. As can be seen, both variants exhibit improvements over $\mathcal{S}_{\text{only}}$, highlighting the effectiveness of individual spatial and temporal divergence removal within *MetaTrans*. Furthermore, the integration of both, as in *MetaTrans*, significantly enhances adaptation performance.

*Table 2.* Ablation study on the Epic-Kitchens dataset.

| Model Variants | P08 → P01 | P08 → P22 | P01 → P08 | P01 → P22 | P22 → P08 | P22 → P01 | Average ↑ |
|---|---|---|---|---|---|---|---|
| Source-only ($\mathcal{S}_{\text{only}}$) | 32.8 | 34.1 | 35.4 | 39.1 | 34.6 | 35.8 | 35.3 |
| MetaTrans_wo_sub | 42.1 | 45.6 | 44.6 | 53.8 | 43.9 | 51.7 | 47.0 |
| MetaTrans_wo_adv | 42.4 | 43.1 | 42.5 | 53.6 | 44.8 | 51.7 | 46.4 |
| MetaTrans | **48.0** | **50.4** | **47.4** | **56.6** | **48.5** | **55.1** | **51.0** |

## B.3. Permutation Invariance Analysis.

We further conduct permutation invariance analysis on the Epic-Kitchens dataset. The two alternatives follow the definition in the main paper, i.e., *MetaTrans_fs_pooling* and *Bi-LSTM_pooling*. Comparison results are presented in Table 3. As can be seen, both variants outperform $\mathcal{S}_{\text{only}}$, yet neither matches the performance of *MetaTrans*.

*Table 3.* Permutation Invariance Analysis on the Epic-Kitchens dataset.

| Model Variants | P08 → P01 | P08 → P22 | P01 → P08 | P01 → P22 | P22 → P08 | P22 → P01 | Average ↑ |
|---|---|---|---|---|---|---|---|
| Source-only ($\mathcal{S}_{only}$) | 32.8 | 34.1 | 35.4 | 39.1 | 34.6 | 35.8 | 35.3 |
| *MetaTrans_fs_pooling* | 42.0 | 41.1 | 43.7 | 50.6 | 40.6 | 50.7 | 44.8 |
| *Bi-LSTM_pooling* | 44.4 | 41.5 | 44.6 | 51.0 | 42.8 | 52.1 | 46.1 |
| MetaTrans | **48.0** | **50.4** | **47.4** | **56.6** | **48.5** | **55.1** | **51.0** |

### B.4. Sensitivity Analysis

*MetaTrans* only contains one importance weight, i.e., $\lambda_1$. Following the common protocol in UVDA, we conduct the grid search from 0.01 to 0.1 with 0.01 as the step on the validation set for the optimal value selection. The sensitivity analysis results are shown in Table 4. As evident, incorporating adversarial loss consistently yields superior results compared to its absence, which shows the necessity of reducing temporal domain divergence in UVDA. Moreover, we find that *MetaTrans* achieves the best result when $\lambda_1$ is 0.03 and 0.07 for $\mathbf{H} \rightarrow \mathbf{U}$ and $\mathbf{U} \rightarrow \mathbf{H}$, respectively. We thus set these values for our experiments.

*Table 4.* Sensitivity analysis on $\lambda_1$.

| $\lambda_1$ | 0 | 0.01 | 0.02 | 0.03 | 0.04 | 0.05 | 0.06 | 0.07 | 0.08 | 0.09 | 0.1 |
|---|---|---|---|---|---|---|---|---|---|---|---|
| $\mathbf{H} \rightarrow \mathbf{U}$ | 93.9 | 94.7 | 96.3 | **99.0** | 96.6 | 95.4 | 94.7 | 96.3 | 95.7 | 93.9 | 94.7 |
| $\mathbf{U} \rightarrow \mathbf{H}$ | 82.5 | 85.3 | 85.3 | 86.1 | 90.6 | 88.1 | 89.2 | **92.2** | 88.7 | 86.1 | 90.6 |

### B.5. Additional *RGRA* Results

In Eq. (8) of the main paper, a strategy of searching one by fixing the other weights is used in the definition of *RGRA*. Herein, we explore the greedy search strategy. Consequently, *RGRA* is formulated in the following:

$$RGRA = \frac{A_{opt} - A_{\mathcal{S}_{only}}}{A_{\mathcal{T}_{sup}} - A_{\mathcal{S}_{only}}} * \frac{1}{10^{(N_{loss}-1)}}. \tag{26}$$

The corresponding comparison results of *RGRA* are shown in Table 5. As can be seen, the *RGRA* of *MetaTrans* remains the same as presented in Table 3 of the main manuscript as only one loss is used in *MetaTrans*. However, for other methods employing multiple losses, the use of the greedy search strategy further diminishes the return over investment of the method. Particularly, *TranSVAE* emerges as the least efficient baseline due to the largest number of losses utilized. This series of experiments further corroborates the exceptional superiority of *MetaTrans* over existing baselines in terms of efficiency.

*Table 5.* *RGRA* (as in Eq. (26)) comparison results (%) on UCF-HMDB and Epic-Kitchens.

| Method | $N \downarrow$ | P08 → P01 | P08 → P22 | P01 → P08 | P01 → P22 | P22 → P08 | P22 → P01 | Average↑ | U → H | H → U | Average↑ |
|---|---|---|---|---|---|---|---|---|---|---|---|
| DANN | 2 | 1.57 | 0.84 | 1.34 | 1.14 | 1.88 | 2.23 | 1.49 | 0.38 | −0.86 | −0.06 |
| ADDA | 2 | 0.83 | 0.27 | 0.42 | 0.69 | 0.67 | 1.99 | 0.80 | −0.75 | −0.43 | −0.63 |
| TA3N | 3 | 0.04 | 0.11 | 0.25 | 0.15 | 0.24 | 0.30 | 0.18 | 0.08 | 0.22 | 0.13 |
| MM-SADA | 3 | 0.39 | 0.19 | 0.29 | 0.28 | 0.33 | 0.37 | 0.33 | 0.27 | 0.29 | 0.28 |
| STCDA | 3 | 0.48 | 0.24 | 0.42 | 0.35 | 0.29 | 0.34 | 0.35 | 0.19 | 0.41 | 0.27 |
| CoMix | 3 | 0.32 | 0.23 | 0.15 | 0.25 | 0.34 | 0.48 | 0.30 | 0.43 | 0.63 | 0.50 |
| MixDANN | 3 | 0.49 | 0.30 | 0.37 | 0.20 | 0.38 | 0.39 | 0.36 | −0.19 | −0.28 | −0.22 |
| CLDA | 3 | 0.38 | 0.23 | 0.43 | 0.27 | 0.27 | 0.40 | 0.33 | 0.57 | 0.97 | 0.71 |
| DFRA | 4 | 0.04 | 0.03 | 0.06 | 0.04 | 0.04 | 0.05 | 0.04 | 0.06 | 0.10 | 0.07 |
| TranSVAE | 5 | 0.01 | 0.01 | 0.01 | 0.01 | 0.01 | 0.01 | 0.01 | 0.01 | 0.01 | 0.01 |
| MetaTrans | 2 | **4.87** | **5.51** | **5.56** | **7.11** | **6.21** | **6.84** | **6.02** | **8.11** | **12.59** | **10.35** |

## C. Remarks on *MetaTrans*.

For *MetaTrans*, we want to highlight the following remarks.

- Visual features, with and without positional embeddings, undergo processing by same self-attention blocks to maintain consistency. Essentially, $\mathcal{M}_1$ and $\mathcal{M}_2$ have identical model structure.

- The permutation-invariant nature is pivotal. While there are alternative model architectures to extract static feature representation from sequences, such as LSTM with average pooling, these architectures cannot ensure temporal permutation-invariance.

- *MetaTrans* leverages the robust temporal representation learning capability of Transformer w. positional embedding, and the temporal permutation-invariant nature of a Transformer variant w.o. positional embedding, to learn the temporal feature representation and the static feature representation, respectively. This is different from either the disentangled idea in *TranSVAE* (Wei et al., 2023) that uses information bottleneck to achieve the static and dynamic separation, or separate encoder structures for human and context information in HCT (Lin et al., 2024).

- Note that *TranSVAE* uses a static consistency constraint to approximate permutation invariance, but this constraint is limited to a single random permutation. In contrast, with Theorem 1, the temporal permutation-invariance of *MetaTrans* holds for any permutation.

- *MetaTrans* is more than an application of attention blocks to UVDA. It embodies a streamlined design grounded in a theoretically-justified idea: the effective management of spatial and temporal divergence by the extraction of high-quality static and temporal feature representations, all accomplished with a simple objective function.

## D. Limitation and Future Work.

We explore the limitations of *MetaTrans*, which can point towards promising future directions. Firstly, *MetaTrans* relies on a conventional self-attention architecture with standard positional embeddings. Exploring more advanced attention-based variants might significantly boost its performance. Secondly, alternative temporal permutation-invariant models could exist, and integrating them with our proposed decoupled adaptation approach could be fruitful. Thirdly, *MetaTrans* could be strengthened by considering additional factors such as incorporating multi-modality inputs, leveraging advanced backbone features like videomae, and evaluating its performance in multi-source adaptation scenarios as well as other video adaptation tasks. These unexplored avenues hold the potential to provide valuable insights for future research.

