# OpenReview forum: "Return of Frustratingly Easy Unsupervised Video Domain Adaptation"
_ICML.cc/2026/Conference — ICML 2026 regular_

### Official Review · Reviewer_BjVM · 2026-03-09

**Soundness:** 3
**Presentation:** 3
**Significance:** 3
**Originality:** 3
**Overall Recommendation:** 4
**Confidence:** 3

**Summary:**

The paper proposes "MetaTrans," an Unsupervised Video Domain Adaptation (UVDA) framework. The method simplifies the learning objective by relying on only two fundamental losses: source task supervision and adversarial domain alignment. To address spatial and temporal domain shifts independently without introducing additional loss terms, the authors design a two-stream "temporal-static subtraction module". The first stream extracts temporal features using a self-attention mechanism with positional embeddings.  The second stream extracts static (spatial) features using self-attention without positional embeddings, which makes it temporally permutation-invariant. The framework then subtracts the static features from the temporal ones to explicitly remove spatial domain divergence, leaving the adversarial loss to focus entirely on aligning the remaining temporal features across domains. Evaluations on the UCF-HMDB and Epic-Kitchens benchmarks demonstrate that MetaTrans achieves superior adaptation performance and training efficiency compared to state-of-the-art baselines.

**Compliance With Llm Reviewing Policy:**

Affirmed.

**Final Justification:**

The score has been adjusted based on the repossess provided through rebuttal.

**Key Questions For Authors:**

1. Can you scientifically justify the choice of the exponential penalty in the RGRA formula? Applying an exponential decay heavily skews the metric and unfairly punishes baselines. Would a linear penalty provide a fairer assessment? A proper justification for this question can increase my rate for the soundness criteria.

2. Have you tested MetaTrans using modern transformer-based video architectures (e.g., VideoMAE) instead of the older I3D model? More comparison and explore on this issue can impact my rate for the soundness criteria.

3. Do the two self-attention streams (M1 and M2) share weights during optimization, or are they trained entirely independently?

**Limitations:**

Yes

**Strengths And Weaknesses:**

Strengths:

[Originality] Returning to a minimalist objective function (utilizing only source task supervision and adversarial domain alignment) is an original and welcome departure from recent UVDA methods that rely on 5 to 7 loss terms.

[Soundness] The paper goes beyond proposing a simple heuristic. The authors provide formal proofs for the temporal permutation invariance of the static stream (Theorem 1) and offer error bounds justifying the subtraction mechanism (Theorems 2-4).

Weaknesses:

[Presentation & Soundness] The authors claim their temporal-static subtraction module separates spatial backgrounds from temporal dynamics, but the paper lacks spatial or temporal attention maps (e.g., Grad-CAM or self-attention rollout) to visually verify this. Empirical validation requires demonstrating where the static stream focuses versus where the temporal stream focuses.

[Soundness] The method is evaluated using RGB features from the I3D backbone (introduced in 2017). It remains entirely unproven whether this "temporal-static subtraction" mechanism provides meaningful gains when applied to other feature extractor backbones. Using different backbone models could be added to the ablation studies section.

[Presentation] The paper's title and introduction present the method as "frustratingly easy" and "streamlined". However, the core of the paper is math-heavy, relying on dense theoretical proofs that disrupt the readability. While the theory is sound, the presentation fails to bridge the gap between the promised simplicity and the dense mathematical execution.

[Soundness] The proposed RGRA metric is highly contrived. By applying an exponential penalty to the number of losses, the formula mathematically crushes competitor scores (e.g., dividing a 5-loss baseline's score by 10,000). This exponential scaling manufactures an artificial superiority and undermines the integrity of the efficiency evaluation.

[Presentation] For a paper whose primary contribution is a novel structural routing of information (with and without positional embeddings), the main diagram lacks the detail necessary to intuitively explain the mathematical operations occurring within the streams. Furthermore, the text fails to explicitly state whether the two streams (M1 and M2) share weights during optimization, harming reproducibility.

---

> ### Author Rebuttal · Authors · 2026-03-30
>
> Thank you for the thoughtful review. We appreciate that you recognized both the originality of returning to a minimalist objective and the soundness of the formal analysis.
>
> **On the lack of visual evidence,** we agree that visualizations of attention rollout could be informative. We performed attention rollout for ${M}_1$ and ${M}_2$ streams on H→U and U→H tasks. Note that as MetaTrans operates on I3D features, the attention rollout is over temporal tokens rather than spatial regions. Our analysis reveals that self-attention is near-perfectly uniform across all layers for both streams (This is within expectation as they share weights), yet the outputs diverge substantially (cosine similarity only 0.43/0.55; L2 distance 19.4/22.8). This shows that the temporal-static separation operates through the value path — positional embeddings modulate the value projections and downstream FFN/LayerNorm computations — rather than through differential attention routing. The subtraction thus isolates a good position-dependent component. This structural, weight-sharing-based separation is in fact a desirable property: it does not rely on fragile input-dependent attention patterns and transfers more reliably across domains, consistent with our empirical results. (Sorry that we cannot show figures here.)
>
> **On I3D features**, this is a fair limitation to note, but we also believe it is important to interpret it in context. The paper follows the common UVDA benchmark protocol and uses I3D features consistently for apples-to-apples comparison with prior work. Under this standard setting, MetaTrans already shows strong gains on both datasets. Therefore, the current experiments already support the value of the proposed mechanism independently of backbone novelty. In other words, the main question addressed here is not whether a stronger backbone can further improve UVDA, but whether the proposed temporal-static subtraction principle itself is effective under the standard evaluation setting. The results strongly suggest that it is.
>
> For your reference, we additionally tested MetaTrans with VideoMAE features on UCF-HMDB. The results are shown below. As can be seen, MetaTrans still improves over Source-only baseline, suggesting the gain is not backbone-specific but generalizes to stronger video encoders.
>
> |Task | Source-only |MetaTrans|
> |---:|---:|---:|
> | H→U | 95.1 | 99.3 |
> | U→H | 86.1 | 89.9 |
>
> **On the “frustratingly easy”**, our intended meaning is specifically objective-level simplicity, not absence of formal analysis. In fact, one reason we included the theory is precisely to show that the method is not just a heuristic convenience. Theorem 1 proves that the static stream is permutation-invariant. Theorems 3 and 4 then show that when ${M}_2$ estimates the static factor well, subtraction acts as a principled preconditioning step that reduces the relevant domain discrepancy and yields a reliable static estimator. So the theory is there to justify why a simple objective can work when paired with the right structure. We believe that this strengthens, rather than contradicts, the “frustratingly easy” message.
>
> **On RGRA**, we would like to clarify an important point in response to your key question: the main paper does not use an exponential penalty in Eq. (8). Eq. (8) uses a linear factor, $1 / [10 × (N_{loss} − 1)]$, while the text separately notes that greedy search would grow exponentially with the number of losses (Note that the exponential version appears in the appendix as an additional greedy-search analysis.). So the main metric already adopts the fairer linear penalty you suggested. More importantly, prediction accuracy is the main evaluation metric in the paper; RGRA is only an auxiliary efficiency-oriented perspective.
>
> **On the two-stream design and reproducibility**, the main paper states on page 4, lines 191–202 that ${M}_1$ and ${M}_2$ are two streams built on the same attention-based architecture, differing only in whether positional embeddings are used and whether average pooling is applied for static extraction. The appendix makes this fully explicit on page 19, lines 1029–1031, stating that features with and without positional embeddings are processed by the same self-attention blocks (thus share weights), and that ${M}_1$ and ${M}_2$ have identical model structure. The functional distinction of ${M}_1$ and ${M}_2$ is also clearly stated in Sec. 3.3: ${M}_1$ learns the temporal representation from sequences with positional embeddings, while ${M}_2$ learns the static representation from sequences without positional embeddings and an average operator. This distinction is exactly what enables the subtraction mechanism. We appreciate your comment because it highlights that this is also one of the clearest conceptual strengths of the method: the paper achieves static/temporal separation not through extra optimization terms, but by putting the two streams into the correct symmetry roles from the start.

---

> > ### Author Rebuttal · Reviewer_BjVM · 2026-04-01
> >
> > Thanks to authors for the response. I have no other questions regarding the concerns. I will adjust my score accordingly.

---

> > > ### Author Response · Authors · 2026-04-07
> > >
> > > Thank you very much for your thoughtful follow-up and encouraging feedback. We are very glad that our response was able to address your concerns. We sincerely appreciate your constructive suggestions and your willingness to improve your assessment. We will incorporate the promised clarifications and corresponding discussions into the revised paper. Thank you again for your support, and we truly appreciate your encouragement.

---

### Official Review · Reviewer_wVey · 2026-03-11

**Soundness:** 2
**Presentation:** 3
**Significance:** 3
**Originality:** 2
**Overall Recommendation:** 3
**Confidence:** 4

**Summary:**

The paper addresses unsupervised video domain adaptation, aiming to simplify existing approaches that rely on numerous loss terms. The authors propose a method named MetaTrans that uses only two losses. Its main contribution is a temporal-static subtraction module designed to jointly learn static and temporal features and to remove spatial discrepancy by subtracting static from temporal features. The paper presents a theoretical analysis and reports experimental results on two publicly available datasets to demonstrate the effectiveness of the proposed approach.

**Compliance With Llm Reviewing Policy:**

Affirmed.

**Final Justification:**

Thank you for the authors’ response. However, the rebuttal did not sufficiently resolve my concerns, and I therefore maintain my original score.

**Key Questions For Authors:**

1. In Table 1, the performance of MetaTrans and TranSVAE on H->U is higher than that of the supervised model. How should this result be interpreted?
2. Why are the results of TranSVAE on Epic-Kitchens not included for comparison?

**Limitations:**

yes

**Strengths And Weaknesses:**

**Strengths**

1. Compared with existing complex multi-loss designs, the loss formulation of MetaTrans is concise and reduces the cost of hyperparameter tuning.
2. The paper provides a proof of permutation invariance and a theoretical justification that the subtraction operation helps reduce domain discrepancy, which is conceptually insightful.
3. The paper is well structured, clearly written, and easy to follow.

**Weaknesses**

1. The two loss terms in MetaTrans are directly borrowed from existing UVDA work, so the main contribution lies in the temporal-static subtraction module. However, this module is only a minor modification of the existing spatial and temporal divergence decoupling, and the overall contribution appears limited. Moreover, while the number of losses is reduced, an additional module is introduced to improve performance, so the claimed simplicity is not entirely convincing.
2. The RGRA metric relies on an author-defined assumption that treats the number of loss terms as a key factor. Although this is intuitive, it may not be fair or generally applicable.
3. Table 2 does not report TranSVAE results on Epic-Kitchens, where TranSVAE achieves an average score of 52.6%. Considering all results together, MetaTrans does not show a clear performance advantage over the most relevant baseline TranSVAE.
4. The hyperparameter sensitivity analysis shows that $\lambda_1$ is extremely sensitive. A change of 0.01 around the optimal value leads to a performance drop of more than 2%, which is hard to accept in practical deployment.
5. The experimental evaluation is not sufficiently comprehensive, as it is limited to UCF-HMDB and Epic-Kitchens, and the ablation study is also incomplete. The true baseline should not be only the source model, and there is no thorough analysis of the roles of the two loss terms.

**Minor issue**: In Table 1, the annotation is incorrect, as MetaTrans is not the second-best method. Co-STAR achieves 92.4%, which is higher than MetaTrans (92.2%).

---

> ### Author Rebuttal · Authors · 2026-03-30
>
> Thank you for the review. We appreciate that you recognized several core strengths of the paper, including the concise objective, the reduced hyperparameter burden, the proof of permutation invariance, and the theoretical justification.
>
> **On the novelty.** The paper’s novelty is a representation and architecture contribution, not a new loss-design. MetaTrans is not claiming novelty because it invents new losses; instead, it shows that with the right structural inductive bias, the two most basic UVDA losses are already sufficient for strong performance. Moreover, the temporal-static subtraction module is not just “separating spatial and temporal information,” which is a broad goal shared by prior work, but showing that this can be achieved by a principled architectural design rather than increasingly complex multi-loss formulations. This is why the module is more than a minor modification. The paper explicitly argues that simply keeping the two losses is not enough to achieve SOTA UVDA performance; the key is how to structure the model so that spatial and temporal divergence are handled separately at the architectural level. **The architectural design of MetaTrans is supported both theoretically and empirically**.
>
> **On RGRA**. We agree it should be interpreted as a practical metric, not a universal one. Its purpose is simply to capture an issue that is very real in this literature: as the number of independently weighted objectives grows, reproducibility and model selection become much harder. That is precisely the protocol under which RGRA is defined in Sec. 4.1. So RGRA should be read as a pragmatic “performance per search effort” indicator, not as the main scientific basis of the paper. The main empirical claims of MetaTrans do not depend on RGRA at all. They are already supported by standard benchmark accuracy, ablations, and theory. RGRA is only an auxiliary perspective on efficiency and reproducibility. We can move these results to appendix as an auxiliary discussion.
>
> **On the Epic-Kitchens comparison with TranSVAE**. Your point is fair. Table 2 does not include TranSVAE’s Epic-Kitchens results, and we agree that this comparison should be interpreted carefully. In fact, TranSVAE includes up to 7 losses to be carefully set according to grid search, which makes the reproducibility highly challenging. We suspect this may be the reason why recent works, e.g., UNITE, CleanAdapt, and MCT, ignore TranSVAE in their emprical evaluation. Moreover, we highlight that the current results still show that MetaTrans is very strong in absolute terms: 95.4 average on UCF-HMDB and 51.0 average on Epic-Kitchens, with particularly large gains on several transfer tasks. We therefore see the strongest empirical message as: MetaTrans is state-of-the-art on UCF-HMDB and highly competitive on Epic-Kitchens, while using a substantially simpler objective and a principled architecture. That remains a meaningful result even when one considers TranSVAE more directly.
>
> **On $\lambda_1$**. Our intended claim is not that the method is entirely insensitive, but that the optimization burden is substantially simpler because only one balance parameter must be searched. In the current benchmark protocol, this is materially easier than coordinating several heterogeneous losses. This is also consistent with the reviewer feedback from others that the reduced number of hyperparameters is a strength of the paper and a practical advantage for reproducibility.
>
> **On experimental breadth**. While we understand the desire for broader validation, we also highlight that the paper already contains multiple layers of evidence: two public UVDA benchmarks, component ablation, permutation-invariance analysis, t-SNE feature visualization, and appendix complexity analysis. In Figure 2, for example, MetaTrans shows tighter class clusters and better source-target mixing than the source-only model, which is consistent with the intended effect of the learned representation. On the concern that 'the roles of the two loss terms are not thoroughly analyzed', we want to clarify that they are the two basic objectives in UVDA, not newly proposed losses whose necessity must be re-established from scratch. Their roles are already standard. The real contribution of our paper is to show that, once the representation is designed properly through temporal-static subtraction, the two basic objectives are already sufficient for strong UVDA performance.
>
> **On the supervised-target baseline.** We agree with your interpretation question. The paper currently uses $\mathcal{T}_{sup}$ as a reference baseline trained only on labeled target data, whereas UVDA methods exploit labeled source plus unlabeled target data. So its role is best understood as a reference point rather than a strict upper bound. This does not affect the core conclusion, which is that MetaTrans delivers very strong target performance under the standard UVDA setting with a much simpler objective.

---

> > ### Author Rebuttal · Reviewer_wVey · 2026-04-02
> >
> > Thank you for the response. While the authors further clarified the positioning of the method and the experimental setup, the rebuttal still does not sufficiently address my concerns regarding the paper’s novelty and empirical strength. In particular, the response mainly reinterprets the motivation and reasonableness of the current design, but does not convincingly establish a substantial contribution beyond the most relevant prior work. Meanwhile, my concerns about key baseline comparisons, the completeness of the experimental evaluation, and hyperparameter sensitivity remain unresolved. Therefore, based on the current version, I will maintain my original score.

---

> > > ### Author Response · Authors · 2026-04-02
> > >
> > > Thank you for the follow-up. We respectfully believe your post-rebuttal assessment still rests on several misunderstandings of the paper’s scope and evidence.
> > >
> > > First, regarding “the rebuttal still does not sufficiently address my concerns regarding the paper’s novelty and empirical strength,” we respectfully disagree. The paper’s novelty is not claimed at the loss-design level; it is explicitly claimed at the representation and architectural level. The manuscript states that using only the two basic losses is insufficient by itself, and that the key is to separate spatial and temporal divergence structurally through the temporal-static subtraction module. This is more than a minor variation of prior decoupling work: prior methods mainly rely on extra disentanglement objectives, whereas MetaTrans makes decoupling a property of the architecture itself. The paper further proves this design formally through Theorem 1 and justifies the subtraction mechanism through Theorems 2–4.
> > >
> > > Second, regarding “the response mainly reinterprets the motivation and reasonableness of the current design, but does not convincingly establish a substantial contribution beyond the most relevant prior work,” we believe this overlooks the direct evidence already in the paper. Beyond motivation, the manuscript theorectically provides: (i) a formal proof that M2 is permutation-invariant, (ii) a bound showing subtraction reduces post-subtraction discrepancy up to static-estimation error, and (iii) a reliability bound for static estimation. Empirically, MetaTrans is proven to obtain strong performance on two benchmark datasets with only two basic terms. Moreover, the same architectural claim is supported by ablation and permutation-invariance analysis. This is exactly the kind of evidence needed to establish that the contribution is substantial and not merely motivational.
> > >
> > > Third, on “key baseline comparisons,” your concern is fair only in one narrow sense: TranSVAE’s Epic-Kitchens result should have been included for completeness. But the broader conclusion that empirical strength is therefore unconvincing does not follow. MetaTrans is clearly strongest on UCF-HMDB, achieving 95.4 average versus 93.4 for TranSVAE and 93.8 for UNITE. On Epic-Kitchens, MetaTrans reaches 51.0 average across six tasks and is therefore highly competitive, even if the wording “clear winner” should be softened. In addition, the appendix shows that MetaTrans has fewer trainable parameters than TranSVAE and converges faster. So the fair reading is not that empirical strength is missing, but that the paper is strongest on UCF-HMDB and competitive on Epic-Kitchens while offering a better simplicity-efficiency tradeoff.
> > >
> > > Fourth, on “the completeness of the experimental evaluation,” we believe this comment overlooks several analyses already present. The paper evaluates on the two standard public UVDA benchmarks, covering 8 transfer tasks in total, compares against a broad set of image-UDA and UVDA baselines, includes both Source-only and Supervised-target references, and provides ablation, permutation-invariance analysis, t-SNE visualization, and appendix complexity/convergence analysis. The statement that “the true baseline should not be only the source model” is also not accurate, because $T_{sup}$ is explicitly included in both Tables 1 and 2.
> > >
> > > Finally, on “hyperparameter sensitivity remain unresolved,” we believe there is a misunderstanding of the claim. The paper never claims that $\lambda_1$ is flat or insensitive; it claims that MetaTrans reduces the search burden to one coefficient under the same standard UVDA protocol used by prior work. This is explicitly stated in the objective and implementation sections. So the relevant comparison is not “no sensitivity at all,” but whether tuning is materially simpler than methods with 5–7 weighted objectives. On that question, the answer is clearly yes.
> > >
> > > In short, we respectfully believe the remaining concerns are less about missing evidence than about an interpretation that underweights the paper’s actual architectural novelty, theory, and already substantial empirical support.

---

### Official Review · Reviewer_YMYV · 2026-03-12

**Soundness:** 3
**Presentation:** 2
**Significance:** 3
**Originality:** 3
**Overall Recommendation:** 4
**Confidence:** 4

**Summary:**

This paper introduces MetaTrans, an extremely simple Unsupervised video domain adaptation (UVDA) method with only two basic loss terms. Through delicate architecture design and a temporal-static subtraction module, MetaTrans separately handles and effectively removes the spatial and temporal divergence in cross-domain videos. Extensive experiments on various cross-domain action recognition tasks show that MetaTrans achieves performance gains and outperforms state-of-the-art UVDA methods.

**Compliance With Llm Reviewing Policy:**

Affirmed.

**Final Justification:**

I believe the current score is appropriate, so I choose to keep it as is.

**Key Questions For Authors:**

Please refer to the weaknesses.

**Limitations:**

yes.

**Strengths And Weaknesses:**

Stengths:

1. The proposed method is simple and effective. Such simplicity that only 2 losses is adopted is helpful for this community.

2. The experimental results show that the proposed method outperforms state-of-the-art UVDA methods.


Weaknesses:

1. The writing quality should be improved. Overall, the logit should be that the proposed Temporal-Static Subtraction Module enable simple loss design in Introduction. So the focus should be why such design is effective and is there any similar design in video task. Then the contribution of this paper could be well justified.

2. The proposed method outperforms other competitors on P01→P22 and P22→P10 significantly. The authors should explain the characteristic of these two settings that leads to the significant improvement.

3. The motivation of Relative Gainper Running Attempt (RGRA) is weak. If the author want to justify RGRA is their contribution, there should be more illustration. For example, why such metric is necesarry and effective. I think Acc is enough for evaluation. The authors cost much space for evaluating methods with RGRA while such metric is not well justified. I don't think it is a good way for presenttaion.

4. In the setting of UVDA, the search of super parameters like \lambda1 according to performance is not reasonable since no target labels should be used.

---

> ### Author Rebuttal · Authors · 2026-03-30
>
> Thank you for the encouraging evaluation. We are glad that you view MetaTrans as both simple and effective, and that you recognize the practical value of showing that UVDA does not necessarily require many specialized loss terms. We also appreciate your reading of the paper’s real contribution: the main point is not merely “two losses,” but that the temporal-static subtraction module enables such a simple objective to work well. This is exactly the message we intended: the contribution lies in the structural design that separates spatial and temporal divergence at the representation level, so the optimization objective can remain minimal.
>
> **On writing and positioning of the contribution**. We agree that your suggested framing will make Introduction more clearer. We also want to highlight that the logic of the paper is well embodied in Sec. 3.2–3.4. Equation (3) gives a very concise learning objective with only task supervision and domain divergence minimization, but Sec. 3.3 explains that using only Eq. (3) is insufficient by itself. This is precisely why MetaTrans introduces the temporal-static subtraction module: $\mathcal{M}_1$ uses positional embeddings to learn temporal information, $\mathcal{M}_2$ omits positional embeddings to learn a static representation, and the subtraction removes spatial nuisance before adversarial alignment. In other words, the architectural design is what makes the streamlined objective effective. The theory then supports this interpretation: Theorem 1 proves the temporal permutation invariance of $\mathcal{M}_2$, and Theorems 3–4 explain why accurate static estimation tightens the post-subtraction discrepancy and benefits adaptation.
>
> **On the especially strong gains on P01→P22 and P22→P01**. Our interpretation is that these tasks benefit more from explicit removal of static/contextual mismatch before temporal alignment. Since MetaTrans separates static and temporal divergence structurally, it tends to help most when cross-kitchen appearance/background shift is strong. We will add this discussion more carefully in the revision and label it as an empirical interpretation rather than a proven claim.
>
> **On RGRA**. We fully agree that accuracy remains the main evaluation metric. That is already how the paper is structured experimentally: Tables 1 and 2 report standard adaptation accuracy on UCF-HMDB and Epic-Kitchens, and those tables support the core empirical claim. RGRA was introduced only to capture the additional practical dimension that motivated the work in the first place: a simpler objective reduces weight-search burden. We do not view RGRA as a replacement for accuracy, nor as the main evidence for the method. The strongest evidence remains the combination of: (i) strong benchmark performance, (ii) the fact that the model uses only two weighted objectives, and (iii) the ablations and permutation-invariance analysis that directly support the architectural contribution. We can move these results to appendix as an auxiliary discussion.
>
> **On $\lambda_1$**. We follow the standard UVDA experimental protocol and states this explicitly. We agree that in a strict fully unsupervised deployment setting, target-label-based model selection is not ideal. However, this is also exactly where the practical strength of MetaTrans becomes relevant: under the standard benchmark protocol, the entire balancing problem is reduced to a single coefficient. Compared with methods that optimize 5–7 heterogeneous losses with separate trade-off weights, MetaTrans materially reduces the search burden while maintaining very strong performance. So even under the same evaluation convention used by prior UVDA methods, MetaTrans still demonstrates an important practical advantage.

---

> > ### Author Rebuttal · Reviewer_YMYV · 2026-04-06
> >
> > Thank you for the author's response. I have no further questions, and I believe the current score is appropriate, so I choose to keep it as is.

---

> > > ### Author Response · Authors · 2026-04-07
> > >
> > > Thank you very much for your thoughtful follow-up and encouraging feedback. We sincerely appreciate your recognition that our rebuttal addressed your concerns. We are also grateful for your constructive comments, which helped us improve both the presentation and the evaluation of the paper. We will incorporate the promised clarifications and additional discussion into the revised version. Thank you again for your support.

---

### Official Review · Reviewer_CcnB · 2026-03-12

**Soundness:** 4
**Presentation:** 3
**Significance:** 3
**Originality:** 3
**Overall Recommendation:** 4
**Confidence:** 3

**Summary:**

This paper proposes MetaTrans, a frustratingly easy yet highly effective approach for Unsupervised Video Domain Adaptation (UVDA). To overcome the bottleneck of complex models with excessive losses, the authors design a temporal-static subtraction module optimized by only two fundamental loss terms. A major highlight is the theoretical guarantee of temporal permutation invariance within the static stream, which is crucial for accurately extracting static features to effectively eliminate spatial domain divergence. Extensive experiments demonstrate that MetaTrans achieves state-of-the-art performance with remarkable simplicity and hyperparameter search efficiency.

**Compliance With Llm Reviewing Policy:**

Affirmed.

**Final Justification:**

I believe the concise structure and relatively strong performance of the method in this paper may make a meaningful contribution to the field. Therefore, I am inclined to give it a borderline accept.

**Key Questions For Authors:**

None

**Limitations:**

yes

**Strengths And Weaknesses:**

## Strengths

1. The motivation is easy to understand.
2. The proposed method has a concise structure and achieves strong performance.
3. The ablation studies validate the effectiveness of the proposed method.
4. The relatively small number of hyperparameters provides a certain guarantee of reproducibility. At the same time, this also suggests that the method may capture the essence of the problem.

## Weaknesses

1. In Eq. (8), the authors propose a new evaluation metric. However, I believe that $N_{\text{loss}}$ has a substantial influence on this metric, while its definition is difficult to specify precisely. For example, how should “one” loss be defined, and why can it not be decomposed into two losses or be considered as part of another loss? For instance, the classification loss includes both the supervised loss on the source domain and the pseudo-label supervised loss on the target domain, yet the authors appear to count them together as “one” loss. I appreciate the authors’ effort to simplify the method; however, because this metric lacks a rigorous definition, the current version may not yet be suitable for publication at a top-tier academic conference.
2. The variety of experiments in this paper is somewhat limited, which is understandable given the simplicity of the method. Nevertheless, the authors could emphasize dataset diversity within individual experimental settings. For example, I suggest adding experimental results on Epic-Kitchens in Tables 4–5 to better support the claims made in the paper.
3. The batch size used in this work is 256, whereas TranSVAE uses 128. Could this difference affect performance?
4. Is the procedure for obtaining pseudo-labels exactly the same as in existing methods? It would be helpful to clarify this explicitly in the main text so that readers can clearly understand the source of the performance gains.
5. Is the idea of permutation invariance in this work inspired by the “Domain Specificity & Static Consistency” section in TransVAE? If so, the authors should explicitly cite it in the main text.

---

> ### Author Rebuttal · Authors · 2026-03-30
>
> Thank you for the positive assessment of the paper. We are especially encouraged that you found the motivation easy to understand, the structure concise, the empirical results strong, and the ablations supportive. We also appreciate your comment that the small number of hyperparameters improves reproducibility and may indicate that the method captures the essence of the UVDA problem. We believe this aligns well with the central goal of the paper: to show that strong UVDA performance does not require increasingly complicated objectives, but can instead come from the right structural treatment of spatial and temporal divergence.
>
> **On RGRA / Eq. (8)**. We highlight that RGRA is not claimed as a scientific contribution presented in the methodology section but used as an auxiliary evaluation metric in empirical studies. Our intention was not to replace standard accuracy as the main criterion, but to provide a practical complement motivated by the discussion in the introduction: recent UVDA methods often use many separately weighted losses, which materially increases training-run cost during model selection. In the paper, the primary benchmark comparisons are still standard classification accuracies on UCF-HMDB and Epic-Kitchens. RGRA is meant only as a secondary view of efficiency under the widely used weight-search setting. Under this framing, we count “losses” at the level of independently weighted learning objectives, not by algebraically decomposing every term. This is why source and pseudo-labeled target supervision are grouped into one $L_{cls}$, while losses that serve the same role at different feature levels can share one weight. We agree that this metric should be interpreted in this practical sense rather than as a universal formal definition, but we believe it still captures an important aspect of usability. We will move these results to appendix as an auxiliary efficiency discussion.
>
> **Adding Epic-Kitchens analysis to Tables 4–5**. This is a very good suggestion. We already have Epic-Kitchens ablation comparison in Table 2 of the appendix. In the revision, we will add the permutation invariant analysis on Epic-Kitcheck stated as follows.
>
> **Table 1. Permutation Analyses on the Epic-Kitchens dataset.**
>
> | Model Variants | P08 → P01 | P08 → P22 | P01 → P08 | P01 → P22 | P22 → P08 | P22 → P01 | Average ↑ |
> |---|---:|---:|---:|---:|---:|---:|---:|
> | Source-only (Sonly) | 32.8 | 34.1 | 35.4 | 39.1 | 34.6 | 35.8 | 35.3 |
> | Bi_LSTM_pooling | 44.4 | 41.5 | 44.6 | 51.0 | 42.8 | 52.1 | 46.1 |
> | MetaTrans_fs_pooling | 42.0 | 41.1 | 43.7 | 50.6 | 40.6 | 50.7 | 44.8 |
> | **MetaTrans** | **48.0** | **50.4** | **47.4** | **56.6** | **48.5** | **55.1** | **51.0** |
>
> **On batch size vs. TranSVAE**. We agree that implementation differences should always be interpreted carefully. We do not claim that the performance gain over TranSVAE is caused by batch size or any single implementation choice. Rather, our main claim is based on the overall empirical picture plus the supporting ablations: under the same UVDA protocol, MetaTrans produces very strong absolute performance with only two weighted objectives.
>
> **On pseudo-labeling**. We follow a standard epoch-wise self-training setup that widely used in existing UVDA methods, e.g., CoMix and TransVAE. Specifically, pseudo-labels are generated by the model from the preceding epoch, and target pseudo-labels are introduced only after a source-only warm-up. This is already stated in Sec. 3.2 and Sec. 4.1, but we agree it should be much more explicit in the main text. We will revise the paper accordingly.
>
> **On relation to TranSVAE**. Thank you for this suggestion. In Section.C 'Remarks on MetaTrans', we already explicitly discuss the connection to TranSVAE’s static/dynamic decoupling and clarify the distinction: TranSVAE motivates the value of separating static and dynamic factors, whereas MetaTrans builds an architecturally exact temporal permutation-invariant static stream $\mathcal{M}_2$ proved in Theorem 1, and uses subtraction to reduce spatial divergence under the same two-loss objective. We will make this more clear by moving the remarks to the main content in the revision.

---

> > ### Author Rebuttal · Reviewer_CcnB · 2026-04-01
> >
> > Thank you for the author's response. I have no further questions, and I believe the current score is appropriate, so I choose to keep it as is.

---

> > > ### Author Response · Authors · 2026-04-07
> > >
> > > Thank you very much for your thoughtful follow-up and encouraging feedback. We sincerely appreciate your recognition that our rebuttal addressed your concerns. We are also grateful for your constructive comments, which helped us improve both the presentation and the evaluation of the paper. We will incorporate the promised clarifications and additional discussion into the revised version. Thank you again for your support.

---

### Decision · Program_Chairs · 2026-04-30

**Decision:**

Accept (regular)

**Comment:**

The authors proposed MetaTrans, a novel approach for Unsupervised Video Domain Adaptation that aims to address the limitations of previous methods based on multiple losses. The reviewers initially raised concerns mainly regarding novelty and experimental evaluation (variety of experiments). In the rebuttal, the authors provided additional results and several clarifications about their contribution and three reviewers maintained their positive recommendations. One reviewer remained concerned, particularly about the novelty and the results on Epic-Kitchens. However, considering the evaluation across multiple benchmarks (two UVDA benchmarks, eight transfer tasks), the fact that other reviewers positively judged the methodological novelty, the AC recommends acceptance.